# Genome-wide association analysis of plasma lipidome identifies 495 genetic associations

Linda Ottensmann [1] ✉, Rubina Tabassum[1], Sanni E. Ruotsalainen [1], Mathias J. Gerl [2], Christian Klose [2], Elisabeth Widén [1], FinnGen*, Kai Simons[2], Samuli Ripatti [1,3,4] & Matti Pirinen [1,3,5] ✉

The human plasma lipidome captures risk for cardiometabolic diseases. To discover new lipid-associated variants and understand the link between lipid species and cardiometabolic disorders, we perform univariate and multivariate genome-wide analyses of 179 lipid species in 7174 Finnish individuals. We fine-map the associated loci, prioritize genes, and examine their disease links in 377,277 FinnGen participants. We identify 495 genome-trait associations in 56 genetic loci including 8 novel loci, with a considerable boost provided by the multivariate analysis. For 26 loci, fine-mapping identifies variants with a high causal probability, including 14 coding variants indicating likely causal genes. A phenome-wide analysis across 953 disease endpoints reveals disease associations for 40 lipid loci. For 11 coronary artery disease risk variants, we detect strong associations with lipid species. Our study demonstrates the power of multivariate genetic analysis in correlated lipidomics data and reveals genetic links between diseases and lipid species beyond the standard lipids.

Cardiovascular disease (CVD) is the leading cause of mortality and morbidity worldwide[1] with an estimated heritability of about 50%[2]. Plasma lipids, routinely measured via high-density lipoprotein cholesterol (HDL-C), low-density lipoprotein cholesterol (LDL-C), triglycerides (TG), and total cholesterol (TC), are established risk factors for CVD[3]. The modern efficient lipidomics technologies have extended considerably our understanding of the variability and width of circulating lipids. Lipid species including, for example, Cholesterol esters (CE), Ceramides (CER), Diacylglycerols (DAG), Lysophosphatidylcholines (LPC), Phosphatidylcholines (PC), Phosphatidylcholine-ether (PCO), Phosphatidylethanolamines (PE), Phosphatidylethanolamine-ethers (PEO), Sphingomyelins (SM) and Triacylglycerols (TAG) potentially improve CVD risk assessment over standard lipids[4–14]. Eventually, a better understanding of biological factors underlying lipid metabolism and its connection to CVD

pathophysiology may also provide new treatment options for CVD.

Genome-wide association studies (GWAS) have revolutionized our understanding of genetic variation behind lipid levels[15–35]. With growing sample sizes, more efficient genetic fine-mapping methods, and the use of population isolates like Finland, several likely causal coding variants and genes have been identified. For example, recently reported stop-gained variants in *CD36*, *ANGPTL8*, and *PDE3B* provide potential targets for the next generation of lipid-lowering medications[20,36].

Very large genetic studies have already been conducted for the standard lipids. For example, the multi-ethnic meta-analysis from the Million Veteran Program study, with a sample size of >600,000 participants, identified 306 loci associated with the standard lipids[19] and a multi-ethnic meta-analysis in 1.65 million individuals identified 941 loci[20]. Despite much smaller sample sizes of lipidome GWAS, they have

[1]Institute for Molecular Medicine Finland, HiLIFE, University of Helsinki, Helsinki, Finland. [2]Lipotype GmbH, Dresden, Germany. [3]Department of Public Health, Clinicum, Faculty of Medicine, University of Helsinki, Helsinki, Finland. [4]Broad Institute of the Massachusetts Institute of Technology and Harvard University, Cambridge, MA, USA. [5]Department of Mathematics and Statistics, University of Helsinki, Helsinki, Finland. *A list of authors and their affiliations appears at the end of the paper. ✉e-mail: linda.ottensmann@helsinki.fi; matti.pirinen@helsinki.fi

identified new lipid-associated genetic variants and provided insights into the genetic architecture of lipid metabolism and cardiometabolic diseases. Additionally, the high-dimensional and correlated structure of the lipidome[27] can be utilized in a multivariate framework[28] to increase statistical power to identify new genetic associations but, to our knowledge, such analyses have not been reported so far.

Here we report univariate and multivariate GWAS of 179 lipid species from 13 lipid classes in 7174 Finnish individuals from the GeneRISK cohort, followed by a phenome-wide association study (PheWAS) of the identified lipid-associated genetic loci in 377,277 biobank participants of the FinnGen study and a colocalization analysis with these endpoints. Altogether, we identify 56 lipid-associated loci including 8 new loci, 2 of which were identified in the univariate GWAS (*AGPAT2, SGPL1*), and 6 were only revealed through the multivariate GWAS (*DTL, STK39, CDS1, KCNJ12, YPEL2, AGPAT3*) demonstrating the gain in statistical power provided by multivariate techniques. Fine-mapping identifies variants with high causal probabilities for 26 loci. We also present detailed lipidomic profiles of known CAD-associated variants. Through the large genome-wide investigation of lipidomic measurements and a new multivariate approach, we provide new lipid-associated loci and new insights into lipid metabolism.

## Results

Using shotgun lipidomics, we detected 179 lipid species belonging to 13 lipid classes covering 4 major lipid categories: glycerolipids, glycerophospholipids, sphingolipids, and sterols (Fig. 1, Supplementary Data 1). A hierarchical clustering based on the absolute pairwise Pearson correlations of the plasma levels of lipids revealed 11 clusters of correlated lipids (Fig. 1, Supplementary Figs. 1 and 2), which were used for the multivariate GWAS. The lipid species included in each cluster and the pairwise correlations between the lipid species in each cluster are provided in Supplementary Data 1 and Supplementary Fig. 3.

### Heritability of lipid species

We estimated the SNP-based heritability of all 179 lipid species using >849k high-quality independent genetic variants. The heritability estimates ranged from 0.0 to 0.45 (Fig. 2 and Supplementary Data 1). Sphingomyelins (SMs) had the highest estimated median heritability (median = 0.35, range = 0.18–0.40) followed by Ceramides (Cer) (median = 0.34, range = 0.05–0.36). Phosphatidylcholine-ethers (PCO) showed the smallest median heritability (median = 0.12, range = 0–0.32) preceded by Phosphatidylethanolamines (PEO) (median = 0.13, range = 0.08–0.14). The lipids containing long-chain polyunsaturated fatty acids (PUFA) (C20:4, C20:5, and C22:6 acyl chains) had slightly higher median heritability (median = 0.27, range = 0–0.45) compared with the other lipid species (median = 0.23, range = 0–0.40). PC 18:0;0_20:4;0 had the highest heritability (0.45, SE = 0.05) of all lipid species followed by CE 20:4;0 (0.44, SE = 0.05). The heritability estimates for lipid species grouped by lipid classes, lipid categories, and PUFA acyl chains are shown in box plots in Supplementary Fig. 4.

### Univariate and multivariate GWAS

We performed univariate GWAS for 179 lipid species and multivariate GWAS for 11 clusters using ~11.3 M high-quality genetic variants with minor-allele frequency (MAF) > 0.002. In the univariate GWAS of 179 lipid species, we identified 26,969 variant-lipid associations at the Bonferroni-corrected significance (BFS) threshold ($P < 7.35e-10$) after a correction for 68 principal components explaining 90% of the variance in the lipidome. The multivariate GWAS of 11 clusters revealed 13,157 variant-cluster associations at BFS ($P < 4.55e-9$). The genomic inflation factors for the univariate and multivariate GWAS ranged between 0.99 and 1.14 (Supplementary Data 2). The Manhattan plots for the lipid classes and multivariate analyses are shown in Supplementary Figs. 5 and 6.

To define independent loci across the lipid species and clusters, we first identified lead variants, individually for each univariate ($N = 179$) and multivariate ($N = 11$) trait, iteratively as the variant with the lowest $P$-value. Then the ±1.5 Mb regions around the lead variants were defined as lipid-associated genomic regions (GWAS regions). A total of 495 BFS GWAS regions (357 and 138 from the univariate and multivariate GWAS respectively) were identified. We identified a set of the most probable causal variants in each GWAS region through fine-mapping and considered each of them as representing a single association signal. We merged the identified signals that were in linkage disequilibrium (LD; $r^2 \geq 0.1$) and combined the overlapping regions across all the 190 traits to form a non-overlapping set of lipid-associated loci. Through this process, described in detail in Methods, Supplementary Figs. 7 and 8 for the locus *LPL* as an example, we identified 98 signals (Supplementary Data 2) located in 56 independent loci across all 190 traits (Supplementary Fig. 9, Table 1). The number of associated loci per lipid species correlated positively with the estimated heritability (adjusted $r^2 = 0.3125$, $P = 2.5e-16$), (Supplementary Fig. 10). We identified 29 additional loci that were associated with lipid

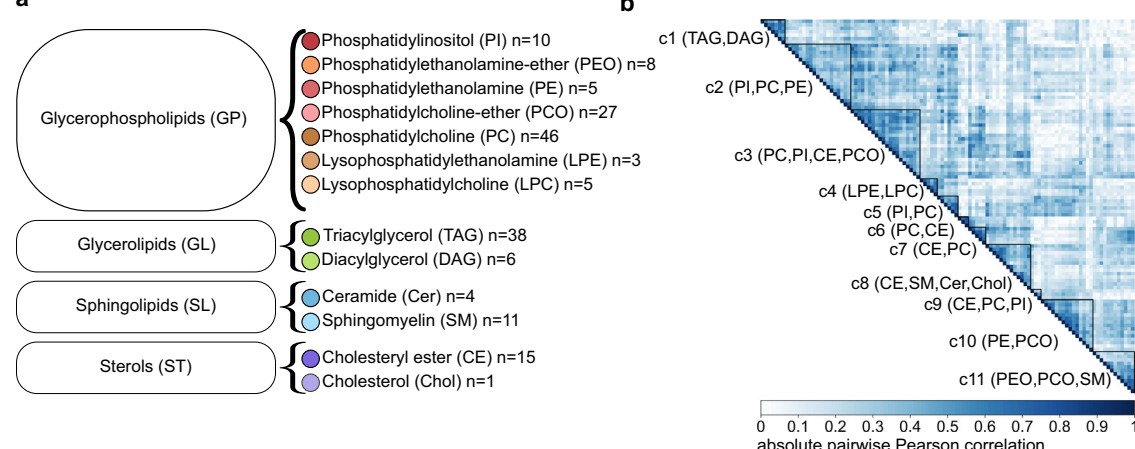

**Fig. 1 | Details of lipid species measured in the GeneRISK cohort. a** The 179 lipid species belong to 13 lipid classes and 4 categories. The lipid class colors are identical to those used in the other figures. **b** Heatmap of the absolute pairwise Pearson correlations between the lipid species included in the 11 clusters of the multivariate GWAS. The clusters are marked by black lines and labeled by the included lipid classes. The members of each cluster are listed in Supplementary Data 1.

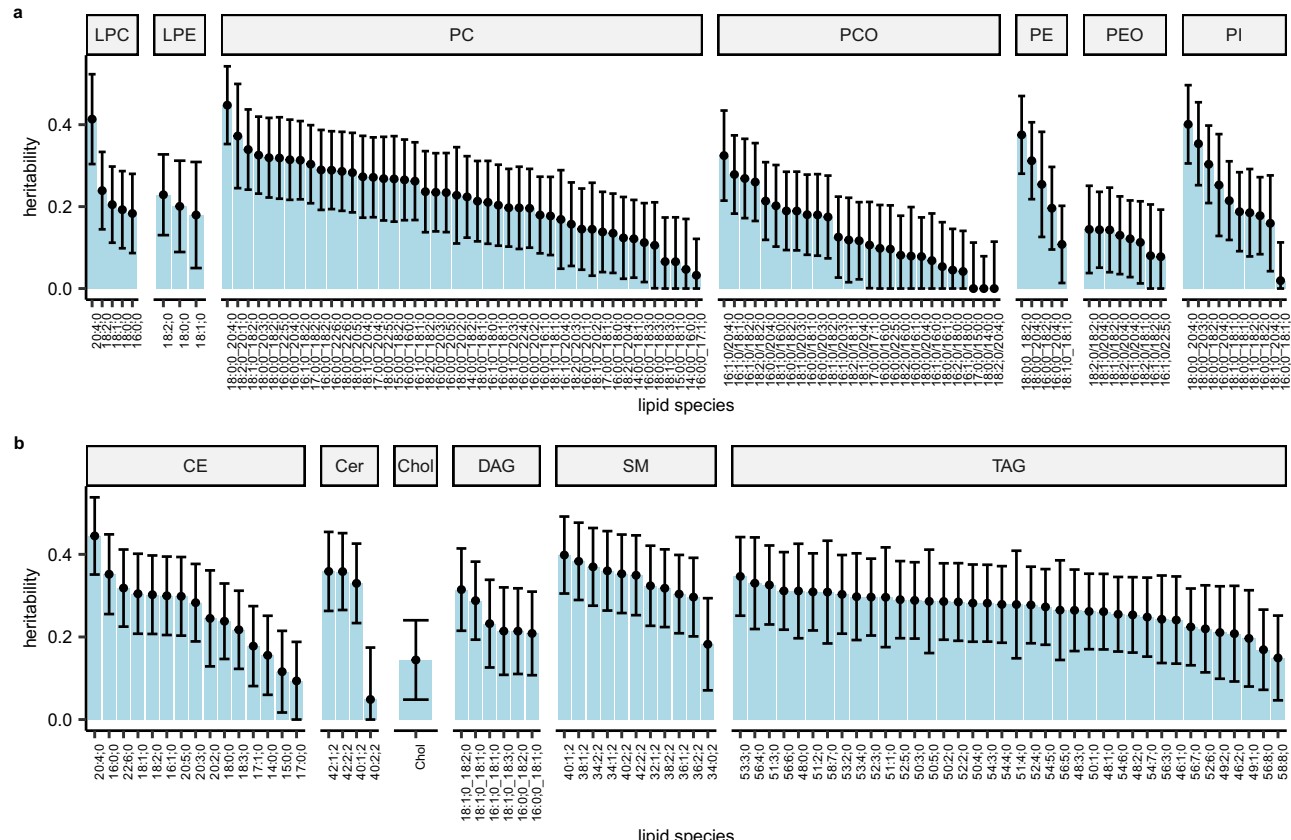

**Fig. 2 | Heritability estimates of lipid species. a** Glycerophospholipids, **b** Glycerolipids, Sphingolipids, and Sterols. Data are presented as heritability estimate ±1.96*SE. The lipid species are presented in descending order of the heritability estimates. Heritability estimation was performed using $n = 7174$ biologically independent samples.

species or lipid clusters at genome-wide significance (GWS) but did not reach BFS (Supplementary Table 1).

We provide the GWAS summary statistics for all 330 lead variants and representative variants for all 179 lipid species (Supplementary Data 2). Of the 59,070 variant-lipid associations, 8238 reached the marginal significance threshold of $P < 0.05/85/68 = 8.650519e{-}6$ corrected for the 85 GWS loci and the 68 PCs that together explained 90% of the phenotypic variance.

Further, to improve the interpretation of multivariate associations we applied MetaPhat[37] to identify the lipid species driving the multivariate association. For 61 of all 138 multivariate BFS GWAS regions, a single lipid species was identified to be driving the association and for 47 of the regions, 2–3 driver traits were identified. The driver traits and other MetaPhat results for the associations reaching GWS in the multivariate analysis are listed in Supplementary Data 2.

Next, we compared the findings of the univariate GWAS and multivariate GWAS. Of the 138 BFS GWAS regions identified in the multivariate analysis, 55 regions did not reach BFS in any univariate analysis of the traits included in that multivariate analysis, for any variant in LD with the lead variant ($r^2 > 0.1$) of the multivariate analysis. The multivariate analysis identified 21 loci not found by the univariate analysis. A comparison of the P-values of the lead variants in the 56 loci in the univariate and multivariate GWAS showed that all the loci identified by the univariate GWAS reached BFS in the multivariate GWAS, except *MARC1* which only reached GWS (Fig. 3). We observed much lower P-values in the univariate compared to the multivariate analysis for *PNPLA3* (6e-19 and 3e-9 for TAG 56:6;0 and cluster 1, respectively). TAG 56:6;0 is not contained in any multivariate cluster, which explains the higher P-value in the multivariate analysis.

### New lipid-associated loci

Altogether, the univariate and multivariate GWAS identified 56 lipid-associated loci including 8 novel lipid loci (Table 2) in or near the following genes: *DTL*, *STK39*, *CDS1*, *AGPAT2*, *SGPL1*, *YPEL2*, *KCNJ12*, and *AGPAT3*. All these loci were identified by the multivariate GWAS but only two were also identified in the univariate GWAS (*AGPAT2* and *SGPL1*). The novel lead variants included a missense variant for *AGPAT3*. *AGPAT2* and *AGPAT3* encode enzymes in the 1-acylglycerol-3-phosphate O-acyltransferase family, whose other member *AGPAT1* is known to be associated with standard lipids[17–20] and lipid species (PC, TAG)[26,35]. AGPAT1/2/3 catalyze the conversion of lysophosphatidic acid to phosphatidic acid in the phospholipid and triacylglycerol synthesis. In our data, these regions were associated with PC and TAG species as well as cluster 3. MetaPhat analysis identified PC species to be driving the associations between cluster 3 and the *AGPAT1/2/3* regions. Two of the identified novel lead variants (*CDS1* and *STK39*) are >2-fold enriched in the Finnish population. The highest enrichment was 69-fold at the *CDS1* locus associated with cluster 2, with PI species as drivers of the association. Of note, *CDS1* encodes an enzyme that regulates the synthesis of PI.

We also report novel associations with lipid species for 11 loci that were previously identified in GWAS of standard lipids. These loci include *ANKRD17*, *ELOVL6*, *ERMP1*, *GPAM*, *HNF4A*, *LCAT*, *MARC1*, and *NPC1L1* (Table 1, Supplementary Data 2).

### Fine-mapping of loci

To identify the most probable causal variants in the associated loci, we performed fine-mapping for both the univariate and multivariate GWAS regions. Of the 56 loci, 26 loci had at least one variant with a high (>0.9) posterior inclusion probability (PIP) in an informative 95%

**Table 1 | Loci reaching the Bonferroni-corrected significance level**

| Locus | Chr | Mv trait | Mv lead variant | Mv minP | Uv trait | Uv lead variant | Uv minP | HDL | LDL | TC | TG | LS |
|---|---|---|---|---|---|---|---|---|---|---|---|---|
| *RF00019* | 1 | 4 | rs7529794 | 1e-12 | LPE 18:0;0 | rs7529794 | 4e-13 | | | | X | X |
| *PCSK9* | 1 | 7 | rs11591147 | 3e-23 | SM 34:1;2 | rs11591147 | 2e-19 | | X | X | | X |
| *DOCK7* | 1 | 2 | rs3913007 | 4e-28 | PI 18:0;0_20:4;0 | rs1168104 | 6e-25 | X | X | X | X | X |
| *AC105942.1* | 1 | 5 | rs1265169 | 1e-25 | PC 18:0;0_18:2;0 | rs34396223 | 7e-13 | | | | | X |
| ***DTL*** | 1 | 5 | rs116329252 | 4e-11 | PC 18:0;0_18:2;0 | rs116329252 | 3e-5 | | | | | |
| *MARC1* | 1 | 1 | rs2642442 | 4e-8 | TAG 54:4;0 | rs2642442 | 5e-11 | X | X | X | X | |
| *GALNT2* | 1 | 10 | rs6672758 | 2e-9 | PC O-16:1;0/18:1;0 | rs6672758 | 4e-10 | X | | | X | X |
| *APOB* | 2 | 7 | rs1367117 | 4e-11 | CE 16:0;0 | rs1367117 | 3e-9 | X | X | X | X | X |
| *GCKR* | 2 | 3 | rs1260326 | 1e-17 | TAG 50:4;0 | rs1260326 | 4e-22 | X | X | X | X | X |
| *ABCG8* | 2 | 9 | rs4245791 | 3e-32 | CE 20:2;0 | rs4245791 | 9e-32 | | X | X | X | X |
| ***STK39*** | 2 | 8 | rs143928330 | 3e-10 | SM 34:2;2 | rs143928330 | 1e-5 | | | | | |
| *AC021074.3* | 3 | 2 | rs12638256 | 1e-39 | PI 18:0;0_18:1;0 | rs12638256 | 5e-20 | | X | X | X | X |
| *ANKRD17* | 4 | 8 | rs187918276 | 4e-15 | SM 40:1;2 | rs187918276 | 3e-17 | X | X | X | X | |
| ***CDS1*** | 4 | 2 | rs191460656 | 2e-9 | PI 16:0;0_18:2;0 | rs191460656 | 3e-5 | | | | | |
| *ELOVL6* | 4 | 3 | rs5022521 | 5e-33 | CE 16:1;0 | rs5022521 | 3e-7 | X | | | X | |
| *SMIM13* | 6 | 6 | rs9468401 | 6e-20 | CE 22:6;0 | rs9468401 | 2e-5 | | | | | X |
| *AGPAT1* | 6 | 3 | rs1061808 | 4e-31 | TAG 50:1;0 | rs1061808 | 4e-9 | X | X | X | X | X |
| *PEX6* | 6 | 6 | rs9462860 | 5e-13 | PC 18:0;0_22:6;0 | rs4987173 | 3e-9 | X | X | | X | X |
| *NPC1L1* | 7 | 8 | rs41279633 | 9e-11 | CE 18:0;0 | rs17725246 | 4e-8 | | X | X | | |
| *MLXIPL* | 7 | 8 | rs3812316 | 9e-12 | DAG 18:1;0_18:2;0 | rs13235543 | 9e-11 | X | | X | X | X |
| *AC022784.1* | 8 | 3 | rs4841133 | 4e-13 | PC 18:0;0_18:2;0 | rs4841133 | 9e-8 | X | X | X | X | X |
| *LPL* | 8 | 1 | rs3916027 | 2e-10 | TAG 56:6;0 | rs10105606 | 2e-12 | X | X | X | X | X |
| *ERMP1* | 9 | 8 | rs142911112 | 1e-20 | SM 32:1;2 | rs140094646 | 1e-8 | | X | X | | |
| *ABO* | 9 | 8 | rs977371848 | 2e-10 | CE 18:0;0 | rs992108547 | 7e-14 | X | X | X | | X |
| ***AGPAT2*** | 9 | 3 | rs2236514 | 2e-12 | PC 16:0;0_22:5;0 | rs2236514 | 4e-12 | | | | | |
| *JMJD1C* | 10 | 8 | rs10822163 | 4e-9 | Cer 42:2;2 | rs10822163 | 4e-6 | X | X | | X | X |
| ***SGPL1*** | 10 | 8 | rs12763964 | 6e-17 | Cer 42:2;2 | rs12763964 | 2e-10 | | | | | |
| *PKD2L1* | 10 | 3 | rs603424 | 4e-26 | PC 16:1;0_18:1;0 | rs603424 | 1e-11 | | X | | | X |
| *GPAM* | 10 | 2 | rs7096937 | 1e-13 | PI 18:0;0_20:4;0 | rs7096937 | 1e-5 | X | X | X | X | |
| *PNLIPRP2* | 10 | 3 | rs4751995 | 9e-12 | PC 16:0;0_18:2;0 | rs4751995 | 1e-6 | X | X | X | | X |
| *FADS2* | 11 | 7 | rs174562 | 1e-783 | PC 18:0;0_20:4;0 | rs174537 | 1e-438 | X | X | X | X | X |
| *CPT1A* | 11 | 3 | rs2229738 | 9e-18 | CE 20:4;0 | rs2229738 | 7e-11 | X | | | X | X |
| *RN7SL786P* | 11 | 3 | rs10160784 | 2e-14 | PC 18:1;0_20:2;0 | rs656095 | 4e-9 | X | | X | | X |
| *ZPR1* | 11 | 3 | rs964184 | 3e-42 | TAG 54:4;0 | rs964184 | 9e-39 | X | X | X | X | X |
| *SOAT2* | 12 | 8 | rs11170421 | 7e-35 | CE 18:0;0 | rs2280696 | 4e-24 | | | | | X |
| *HNF1A* | 12 | 8 | rs1169306 | 6e-22 | SM 38:2;2 | rs1169306 | 4e-12 | X | X | X | | X |
| *AL161670.1* | 14 | 8 | rs7157785 | 2e-197 | SM 32:1;2 | rs7157785 | 3e-95 | | X | X | X | X |
| *LIPC* | 15 | 2 | rs10468017 | 2e-126 | PE 16:0;0_20:4;0 | rs2043085 | 4e-104 | X | X | X | X | X |
| *NTAN1* | 16 | 3 | rs1136001 | 1e-52 | CE 20:3;0 | rs1135999 | 2e-36 | X | | | X | X |
| *CETP* | 16 | 3 | rs17231506 | 1e-33 | PC 16:0;0_18:2;0 | rs17231506 | 8e-15 | X | X | X | X | X |
| *LCAT* | 16 | 7 | rs4986970 | 1e-16 | CE 20:4;0 | rs4986970 | 2e-4 | X | | X | | |
| *GLTPD2* | 17 | 8 | rs79202680 | 1e-79 | SM 40:1;2 | rs79202680 | 4e-60 | | X | X | X | X |
| ***KCNJ12*** | 17 | 7 | rs6587148 | 2e-9 | PC 16:1;0_20:4;0 | rs6587148 | 2e-6 | | | | | |
| ***YPEL2*** | 17 | 11 | rs149807191 | 2e-9 | PC O-16:0;0/20:4;0 | rs149807191 | 7e-8 | | | | | |
| *ABHD3* | 18 | 2 | rs181026394 | 5e-38 | PC 14:0;0_18:2;0 | rs181026394 | 7e-20 | | | | | X |
| *SMUG1P1* | 18 | 2 | rs1540037 | 1e-15 | PI 18:1;0_18:1;0 | rs1540037 | 8e-17 | X | X | X | | X |
| *CERS4* | 19 | 8 | rs2336171 | 2e-189 | SM 38:2;2 | rs2336171 | 6e-53 | X | X | X | X | X |
| *TM6SF2* | 19 | 8 | rs190121281 | 3e-24 | TAG 56:6;0 | rs189452885 | 2e-15 | | X | X | X | X |
| *APOE* | 19 | 8 | rs7412 | 2e-65 | CE 18:2;0 | rs7412 | 8e-36 | X | X | X | X | X |
| *SPHK2* | 19 | 8 | rs61751862 | 6e-16 | SM 34:2;2 | rs61751862 | 1e-7 | | | | | X |
| *TMC4* | 19 | 2 | rs8736 | 2e-301 | PI 18:0;0_18:2;0 | rs8736 | 5e-107 | | X | | | X |
| *LINC01722* | 20 | 8 | rs438568 | 1e-135 | Cer 42:2;2 | rs364585 | 7e-52 | | X | X | | X |
| *NINL* | 20 | 3 | rs6037125 | 6e-12 | CE 20:3;0 | rs6037125 | 2e-6 | | X | X | | X |

**Table 1 (continued) | Loci reaching the Bonferroni-corrected significance level**

| Locus | Chr | Mv trait | Mv lead variant | Mv minP | Uv trait | Uv lead variant | Uv minP | HDL | LDL | TC | TG | LS |
|---|---|---|---|---|---|---|---|---|---|---|---|---|
| *HNF4A* | 20 | 2 | rs1800961 | 1e-10 | CE 18:3;0 | rs1800961 | 1e-10 | X | X | X | | |
| **AGPAT3** | 21 | 3 | rs62229686 | 7e-17 | PC 16:0;0_22:5;0 | rs62229686 | 5e-9 | | | | | |
| *PNPLA3* | 22 | 1 | rs2294915 | 3e-9 | TAG 56:6;0 | rs738409 | 6e-19 | X | X | X | X | X |

Loci found by previous studies for standard lipids or lipid species are marked with an X. Loci are listed as italic gene names. Novel loci are bolded. Two-sided P-values calculated using a linear-mixed-model (uv) and canonical correlation analysis (mv) are reported.

*mv* multivariate, *uv* univariate, *mv trait* cluster number, *uv trait* lipid species, *lead variant* rsid, *minP* minimum of P-values of mv or uv GWAS, *LS* lipid species.

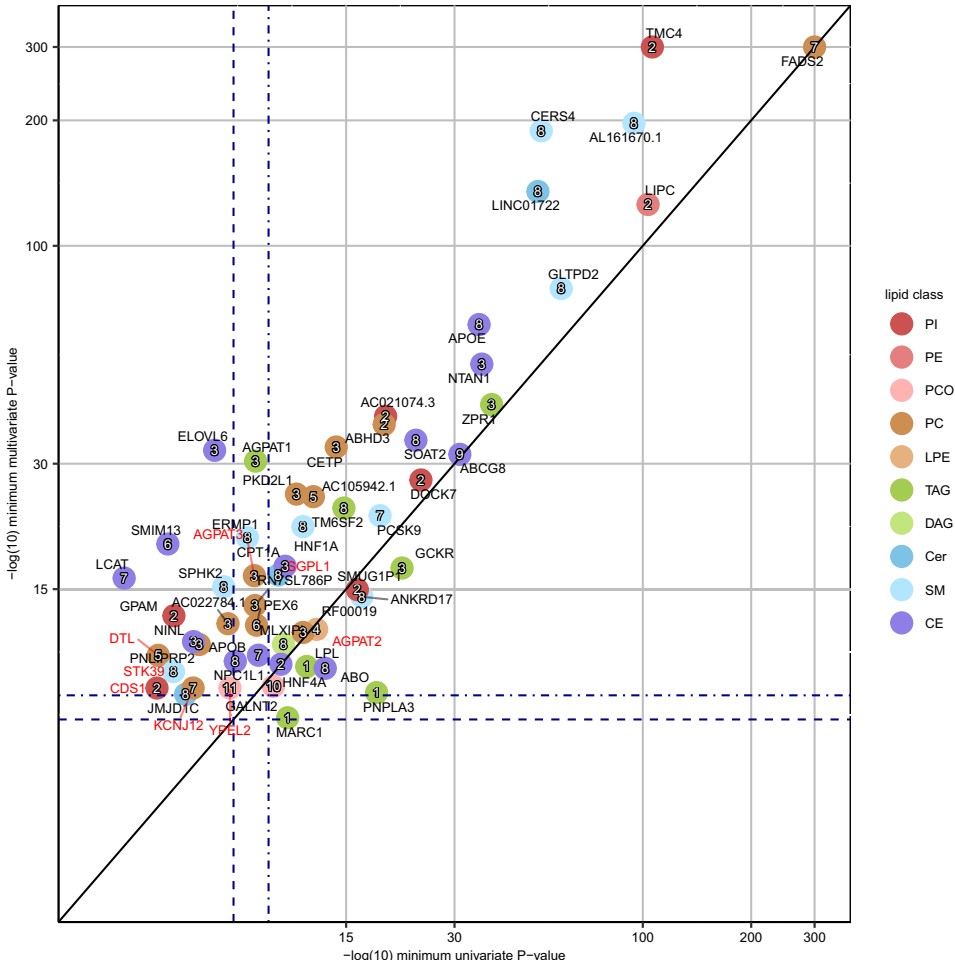

**Fig. 3 | Comparison of the univariate and multivariate *P*-values for 56 lipid-associated loci.** The loci are colored by the lipid class from the univariate analysis and labeled by the cluster number from the multivariate analysis. The *x*-axis shows the *P*-values of the top associated univariate lead variant of the loci. The *y*-axis shows the *P*-values of the top associated multivariate lead variant of the loci. Two-sided *P*-values were calculated using a linear-mixed-model (uv) and canonical correlation analysis (mv). If no variant reached *P* < 5e-8 for the locus in univariate analysis, the minimum univariate *P*-value of the lead variant of the multivariate analysis is shown. Known loci and novel loci are annotated by the locus name in black and red, respectively. Dark blue dashed lines represent the genome-wide significance level (*P* < 5e-8) and dark blue dot-dashed lines represent the multiple testing-corrected significance level (uv: 7.352941e-10, mv: 4.545455e-9). Black line shows the diagonal. The lipid class names are listed in Fig. 1. The axes are capped at 300.

credible set either in univariate or multivariate GWAS (Supplementary Data 3). Altogether, 50 high PIP variants were identified. Variants with a high PIP were found from 13 loci in both the univariate and multivariate analyses (*ABHD3, AGPAT2, APOE, CERS4, GCKR, GLTPD2, HNF4A, LINC01722, LIPC, PCSK9, PKD2L1, SMUG1P1,* and *ZPR1*), from 1 locus (*LPL*) only in the univariate analysis and from 12 loci only in the multivariate analysis (*AGPAT3, CPT1A, DOCK7, ELOVL6, LCAT, MYRF, NPC1L1, SGPL1, SMIM13, SPHK2, STK39, TM6SF2*). Of the 50 variants, 18 variants that reached a PIP > 0.9 in the multivariate analysis had a low

PIP (<0.1) in the univariate analyses. In Supplementary Data 3 the full FINEMAP results are listed and the results for the novel loci are summarized in Supplementary Table 2. The representative variants of the informative credible sets of the BFS univariate and multivariate GWAS regions are plotted by the credible set size against the top posterior inclusion probability (PIP) in Supplementary Fig. 11.

In total, we found 53 variants that affect the molecular function of a protein among the representative variants of credible sets or in high LD (*r*² > 0.95) with them (Supplementary Data 3). These 53 functional

**Table 2 | Novel loci reaching the Bonferroni-corrected significance level**

| Locus | Trait | Chrom | Lead variant | P-value | Function | MAF | Finnish enrichm. |
|---|---|---|---|---|---|---|---|
| *DTL* | cluster 5 (PC 18:0;0_18:2;0) | 1 | rs116329252-A | 4e-11 | intron | 0.04 | 0.87 |
| *STK39* | cluster 8 (SM 34:2;2) | 2 | rs143928330-G | 3e-10 | intergenic | 0.03 | **2.94** |
| *CDS1* | cluster 2 (PI 16:0;0_18:2;0) | 4 | rs191460656-T | 2e-9 | intron | 0.07 | **68.87** |
| *AGPAT2* | PC 16:0;0_22:5;0<br>cluster 3 (PC 16:0;0_22:5;0,<br>PC 16:0;0_22:4;0,<br>PC 18:0;0_22:5;0) | 9 | rs2236514-C | 4e-12<br>2e-12 | intron | 0.35 | 1.00 |
| *SGPL1* | Cer 42:2;2<br>cluster 8 (Cer 42:2;2,SM 34:2;2) | 10 | rs12763964-C | 2e-10<br>6e-17 | intron | 0.21 | 0.86 |
| *KCNJ12* | cluster 7 (PC 16:1;0_20:4;0) | 17 | rs6587148-C | 2e-9 | intron | 0.38 | 0.99 |
| *YPEL2* | cluster 11 (PC O-16:0;0/20:4;0) | 17 | rs149807191-T | 2e-9 | intron | 0.06 | 1.42 |
| *AGPAT3* | cluster 3 (PC 16:0;0_22:5;0,<br>PC 18:0;0_22:5;0) | 21 | rs62229686-T | 7e-17 | missense | 0.04 | 1.44 |

*Locus* locus name identical to gene name from VEP in italic, *trait* lipid species or multivariate cluster (driver trait from metaPhat analysis), *lead variant* rsid-minor allele, *function* variant function from VEP, *Finnish enrichm.* Finnish enrichment calculated as ratio of minor-allele frequencies between our Finnish data and non-Finnish-non-Swedish-non-Estonian European samples in gnomAD v2.1. Bolded if >2. Two-sided *P*-values calculated using a linear-mixed-model (uv) and canonical correlation analysis (mv) are reported.

variants were distributed among 32 of our 56 loci. For the univariate analyses, 34 missense variants and 2 splice region variants were found across 24 loci. For the multivariate analyses, 1 splice acceptor variant (*PNLIPRP2*), 2 splice donor variants (*LILRB3*, *ABHD12*), 1 frameshift variant (*ABHD12*), 1 inframe deletion variant (*NINL*), 38 missense variants and 2 splice region variants were found, distributed across 27 loci. Among the 18 functional variants with PIP > 0.5 in the univariate or multivariate fine-mapping, 9 variants were predicted to be among the top 1% of the most deleterious substitutions (CADD score > 20[38]) and are reaching the GWS threshold in at least one GWAS. These are missense variants for genes *ABHD3*, *APOE*, *APOB*, *G6PC1*, *HNF4A*, *LCAT*, *LIPC*, *LPL*, and *SPHK2*.

The fine-mapping revealed multiple independent signals (Supplementary Data 2) at 24 of the 56 loci including novel signals for well-known lipid genes. For example, for *LIPC*, we found 5 signals represented by variants: rs6493996, rs2043085, rs1077834, rs113298164, and rs201563586, of which the last one is a highly Finnish-enriched missense variant, not in LD with any of the previously reported signals[20,35,39,40].

A comparison to fine-mapping results of the standard lipids in UKB was performed for 45 of our high PIP variants that were directly or through an LD-neighbor ($r^2 > 0.1$) contained in the UKB fine-mapping results (Supplementary Data 3). Of the 45 variants, 15 variants have a CADD score > 10, indicating that the variant is predicted to be among the 10 % of the most deleterious substitutions[38] (Table 3). Of these 15 variants, 8 variants reach a PIP > 0.1 in UKB for at least one standard lipid. The other 7 variants had a PIP < 0.001 for all standard lipids and were not contained in any 95 % credible set in UKB. Of the 7 variants, 3 variants were rare and only reached a PIP > 0.9 in our multivariate analysis. Detailed quality control assessment of these variants is in Supplementary Note 1. Of the 30 variants with a CADD score ≤ 10, 17 variants (or their LD neighbors) reach a PIP > 0.1 in UKB for at least one standard lipid. We observed a lower Pearson correlation between the standard lipids and the lipid species or LCP-phenotypes associated with the variants that had only low PIP in UKB (Supplementary Note 2).

**Gene prioritization**

Next, we prioritized genes in the 98 identified GWS loci first by using functional variants and second by using FOCUS[41], which together prioritized 49 genes (Supplementary Data 4). First, we prioritized genes based on the functional variants that had PIP > 0.5 or that were in high LD ($r^2 > 0.95$) with such variants. Of the 20 prioritized genes, 11 were found both in the univariate and multivariate analysis (*AGPAT3*, *APOE*, *CERS4*, *CPT1A*, *GCKR*, *HNF4A*, *LIPC*, *PCSK9*, *SOAT2*, *SPTLC3*,

*TM6SF2*), 3 only in the univariate analysis (*G6PC1*, *LPL*, *TMC4*) and 6 only in the multivariate analysis (*ABHD3*, *APOB*, *ELOVL2*, *ERMP1*, *LCAT*, *SPHK2*). FOCUS prioritized, at PIP > 0.5, 32 genes of which 17 were found both in the univariate and multivariate analysis (*APOA5*, *AQP9*, *BFAR*, *CETP*, *CNOT3*, *DHX33*, *DOCK7*, *FNDC4*, *GRAMD4*, *LIPG*, *MIB1*, *NOMO1*, *PLEKHH1*, *PNPLA3*, *PPP6R1*, *SCGB2A2*, *SYNE2*), 4 only in the univariate analysis (*APOB*, *APOA1*, *NLRP1*, *SCARB1*) and 11 only in the multivariate analysis (*CCDC86*, *CERS4*, *CNN3*, *DDX49*, *ERMP1*, *GPAM*, *HNRNPM*, *MLEC*, *PRPF19*, *PYGB*, *ZNF506*).

We further assessed gene expression of the prioritized genes in 54 tissues using FUMA[42]. We observed high expression levels in liver for a majority of prioritized genes for both prioritization methods (Supplementary Fig. 12). To assess tissue specificity of prioritized genes FUMA identifies sets of differentially expressed genes (DEG), defined as the gene sets that are more (or less) expressed in a specific tissue compared to all other tissues. The up-regulated DEG sets were significantly enriched ($P \leq 0.05$ corrected for multiple testing) for liver tissue for both gene prioritization methods (Supplementary Fig. 13). The top two enriched gene sets from Gene Ontology biological processes are 'lipid metabolic process' (adjusted $P = 3e-17$) and 'cellular lipid metabolic process' (adjusted $P = 1e-15$) for the functional variant approach, and 'protein containing complex remodeling' (adjusted $P = 1e-9$) and 'lipid homeostasis' (adjusted $P = 2e-9$) for FOCUS. The gene set enrichment results for the prioritized genes are in Supplementary Data 4.

We assessed whether the prioritized genes were included in any gene set from FUMA with the name containing the term lipid. For the functional approach, 3 out of 20 genes were not among the lipid gene sets (*ERMP1*, *G6PC1*, *TMC4*), and for FOCUS, the numbers were 20 out of 32 (e.g. *ZNF506*, *CNOT3*, *GRAMD4*). In total, of the 49 genes, 22 genes were not among FUMA's lipid gene sets.

**Phenome-wide association study (PheWAS)**

To explore the disease relevance of the identified lipid-associated loci, we used data for 953 disease endpoints from 377,277 participants from the FinnGen study. We performed PheWAS for the 264 GWAS lead variants and 287 representative variants of credible sets which were not among the lead variants. We identified 2937 variant-disease associations for variants in 46 GWS loci reaching the *P*-value threshold of $P < 5.24659e-5$ (corresponding to 0.05 corrected for the number of endpoints (953) included in the PheWAS; Supplementary Data 5). Amongst the 9 novel lipid-associated loci, the PheWAS revealed that the cluster 11 associated intronic variant rs149807191 at the *YPEL2* locus associated also with hypertension endpoints (minimum $P = 2e-7$).

**Table 3 | Fine-mapping results**

| Locus | Variant | Function | Gene | CADD | Finnish enrichm. | MAF | Traits (P-value) |
|---|---|---|---|---|---|---|---|
| PCSK9 | rs11591147-T | missense | PCSK9 | 10.4 | **3.10** | 0.033 | CE 18:2;0 (2e-14), 3 SMs: SM 34:1;2 (2e-19), c3 (1e-14), c8 (2e-13) |
| GCKR | rs1260326-T | missense | GCKR | 13.2 | 1.07 | 0.349 | 2 DAGs: DAG 18:1;0_18:2;0 (1e-12), 16 TAGs: TAG 50:4;0 (4e-22), c2 (4e-13), c3 (1e-17) |
| SMIM13 | rs1292311927-T* | splice_region | ELOVL2 | 22.9 | **FIN-specific** | 0.004 | c6 (2e-7) |
| LPL | rs268-G | missense | LPL | 21.3 | 1.01 | 0.023 | 3 TAGs: TAG 54:4;0 (1e-9) |
| LIPC | rs201563586-A* | missense | LIPC | 24.9 | **FIN-specific** | 0.002 | c2 (8e-8) |
|  | rs113298164-T | missense | LIPC | 24.1 | **4.41** | 0.017 | 5 PCs: PC 18:0;0_18:2;0 (1e-12), PC O-16:2;0/18:0;0 (3e-12), 5 PEs: PE 16:0;0_20:4;0 (4e-47), c2 (3e-62), c3 (2e-7), c4 (2e-10), c5 (4e-15), c7 (2e-7), c10 (1e-10) |
| LCAT | rs4986970-T | missense | LCAT | 23.2 | 0.83 | 0.028 | c7 (1e-16) |
| ABHD3 | rs1253048206-G* | intergenic |  | 11.8 |  | 0.017 | c2 (3e-6) |
|  | rs186249276-T* | missense | ABHD3 | 23.7 | **29.64** | 0.004 | c2 (4e-19), c3 (5e-12) |
| APOE | rs7412-T | missense | APOE | 26.0 | 0.56 | 0.053 | 5 CEs: CE 18:2;0 (2e-14), c3 (3e-23), c6 (2e-18), c7 (2e-53), c8 (2e-65), c11 (4e-14) |
|  | rs429358-C | missense | APOE | 16.7 | 1.29 | 0.189 | 2 CEs: CE 20:2;0 (9e-12), c7 (1e-29), c9 (8e-12) |
| SPHK2 | rs61751862-C* | missense | SPHK2 | 22.1 | **2.45** | 0.031 | c8 (6e-16) |
| LINC01722 | rs61738161-A* | missense | SPTLC3 | 18.0 | **2.24** | 0.086 | 3 Cers: Cer 42:2;2 (3e-17), c8 (7e-19) |
| HNF4A | rs1800961-T | missense | HNF4A | 21.4 | 1.41 | 0.052 | 2 CEs: CE 18:3;0 (1e-10), c2 (1e-10) |
| AGPAT3 | rs62229686-T* | missense | AGPAT3 | 16.3 | 1.44 | 0.039 | c3 (7e-17) |

*Locus* gene name in italic, *variant* rsid-minor allele, *variants reaching only low PIP (<0.1) in UKB, *function* variant function from VEP, *gene* gene name from VEP in italic, *MAF* minor-allele frequency, *Finnish enrichm.* Finnish enrichment calculated as ratio of MAF between our Finnish data and non-Finnish-non-Swedish-non-Estonian European samples in gnomAD v2.1. Bolded if >2. Variants not detected outside Finland in gnomAD are marked as FIN-specific.

Variants with a CADD score >10 and a high PIP (>0.9) in GeneRISK are listed. Traits for which the variant reaches a high PIP are listed and, in the case of multiple species of a lipid class, the number of species and the species for which the variant reaches the lowest *P*-value are given. Two-sided *P*-values calculated using a linear-mixed-model (uv) and canonical correlation analysis (mv) are reported.

Figure 4 shows the connection between 9 selected PheWAS endpoints (representing cardiovascular disease, hyperlipidemia, diabetes, metabolic disorders, and neurological disease) and the lipid species and multivariate clusters through shared associated variants. Only the associations reaching the GWS threshold corrected for the number of endpoints ($P < 5.24659e-11 = 5e-8/953$) and simultaneously showing a colocalization (CLPP > 0.01) are illustrated. Of the 179 lipid species, 137 species are included in Fig. 4. We identified 45 instances where links in the loci *FADS2*, *ZPR1*, *CERS4*, *TM6SF2*, and *HNF1A* were detected in PheWAS but not by the colocalization analysis, and they together with the colocalization results for all 953 FinnGen endpoints can be found in Supplementary Data 6. Supplementary Fig. 14 shows the connection of all 45 BFS endpoints, ordered by disease groups, connected with >3 lipid species.

**Coronary artery disease loci associations**
Of the 236 conditionally independent coronary artery disease (CAD) GWS variants at 196 loci[43], 11 reach the BFS threshold $P < 7.35e-10$ for the univariate analysis and 1 additional variant reaches the BFS threshold $P < 4.55e-9$ for the multivariate analysis (Fig. 5). The most widely-associated variant is near *ZNF259*, located in the *BUD13-ZNF259-APOA5-APOA1-SIK3* gene-cluster, which increases the levels of DAGs, PCs, PE 18:0;0_18:2;0, PIs, and TAGs and decreases the level of PC O-16:1;0/18:1;0. This variant is also associated with statin medication ($P = 8e-130$, Beta = +0.19) and disorders of lipoprotein metabolism and other lipidemias ($P = 9e-58$, Beta = +0.18) in FinnGen. We summarized these associations and the 71 associations reaching the significance threshold corrected for multiple testing ($P < 7.352941e-4$ for univariate and $P < 4.545455e-3$ for multivariate analyses) in Supplementary Data 6. Of the 15 CAD variants that were GWS associated with a lipid species or clusters of lipid species, 4 variants with the nearest genes *NAT2* (rs4646249), *LPL* (rs268, rs894211), and *MYH11* (rs12691049) were not located within ±1.5 Mb of the lipid variants

reported by Cadby et al.[35] to be nominally associated with coronary atherosclerosis.

## Discussion
We present a genetic study of plasma lipidome with 7174 participants and 179 lipid species followed by a large-scale PheWAS analysis to reveal new lipid-associated variants and the relationship between lipid species and cardiometabolic disorders. Our study provided several advantages in gaining new information on the genetics of lipid metabolism due to (1) the large sample size of 7174 individuals, (2) the unique genetic background of the Finnish population, (3) high resolution lipidomic measurements, and (4) the multivariate approach. We demonstrate a considerable gain of power from the multivariate analysis of correlated lipid species compared to a commonly used univariate analysis, and expand current knowledge in the field through the analysis of lipidome compared to the standard lipids. We identified variants that were highly associated with both lipid species and disease endpoints, including cardiovascular disease, liver disease, cholelithiasis, diabetes, and lipid disorders.

We found that the heritability estimates of lipid species ranged from 0 to 0.45. Previous studies have reported heritability estimates ranging from 0.10 to 0.54[21,35]. We observed the highest median heritability estimates for the lipid classes SM, and Cer (>0.30). A previous Finnish study reported the highest heritability estimates for Cer (0.39)[21]. In a recent Australian study[35] among the lipid classes included in our study, CE and LPE reached the largest heritability (0.38), and also Cer (0.34) and SM (0.36) were among the most heritable classes. We acknowledge that the differences in heritability estimates between the lipid species may reflect also the differences in measurement accuracy in addition to the differences in the actual heritabilities.

Our sample size is over 3-fold compared to the most recent GWAS on the same lipidome measures (2181 individuals) (Tabassum et al. 2019[21]). This increase is reflected in the number of univariate GWS

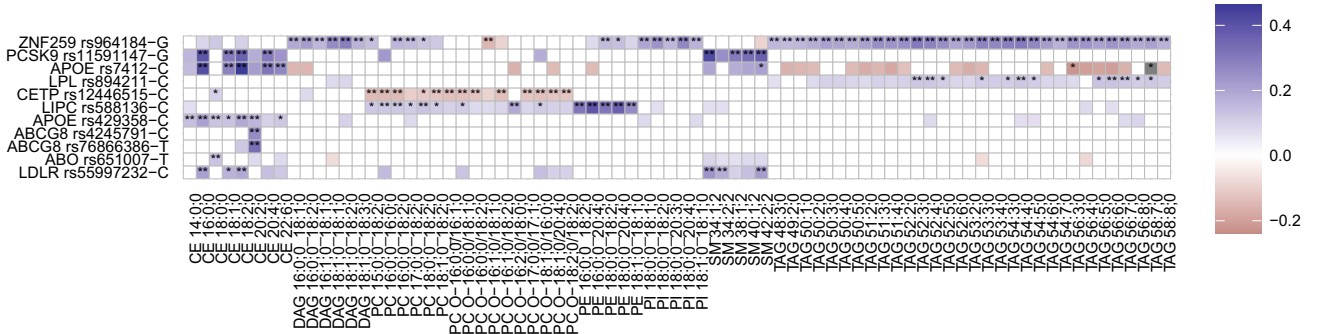

**Fig. 4 | Heatmap of PheWAS associations for selected disease endpoints.** Each entry in the heatmap represents a possible association between a disease group (row) and a lipidome trait (column). The red color indicates that at least one variant among the lead variants or representative variants of the lipidome trait is also associated with the disease at the multiple testing-corrected threshold $P < 5.24659e\text{-}11$ and the lipidome trait and disease colocalize at CLPP > 0.01. Two-sided $P$-values for lipidome traits were calculated using a linear-mixed-model (uv) and canonical correlation analysis (mv). Two-sided $P$-values for disease endpoints were calculated using mixed-model logistic regression. The gray color denotes that no such association is observed. The columns are split by lipid classes. The effective sample size $N_{eff}$ (see the "Methods" section) and the number of loci are given beneath each disease endpoint.

**Fig. 5 | Effect estimates of 11 CAD risk-increasing alleles on lipid species.** Variant ids are defined as rsid-risk-increasing allele. Included are species that reach the multiple testing-corrected threshold of $7.352941e\text{-}10$ for at least one of the variants. Associations reaching the genome-wide-significant ($P < 5e\text{-}8$) or the multiple testing-corrected threshold are indicated by one or two asterisks, respectively. Colored effect estimates are shown for the associations reaching nominal significance corrected for the number of PCs explaining 90% of the variance ($P < 7.352941e\text{-}4$). Two-sided $P$-values for lipid species were calculated using a linear-mixed-model.

findings (68 in our study vs. 35 in Tabassum et al.). Two other recent lipidome studies have been carried out with 5662 Pakistani individuals plus 13,814 British individuals (Harshfield et al. 2021[34]), and 4492 Australian individuals predominantly of European ancestry (Cadby et al. 2022[35]). Even though the sample sizes in lipidome studies are still small compared to the existing GWAS on standard lipids ([19,20]), the high-dimensional lipidome phenotypes complement the standard lipid analyses by identifying new lipid-associated loci, providing a refined picture of the genetic associations and allowing multivariate analyses. Here, we have identified 13 lipid-associated loci that were not captured by any standard lipids study, not even by the largest GWAS of standard lipids with >1.65 million participants[20]. The 10 lipid species associated with these loci are less correlated with the standard lipids than the remaining lipid species (Pearson correlation of 0.41 compared to 0.48, Supplementary Data 1), reiterating that the standard lipids do not completely capture the complex lipid metabolism.

The Finnish genetic background of our study population provides a unique opportunity to discover variants that are enriched in the Finnish population but extremely rare outside of Finland, and to identify new independent signals in known lipid loci. We identified two new lipid-associated loci that are enriched in the Finnish population. These two enriched loci and the other 6 new loci reach $P$-values between 9e-6 and 2e-3 for lipids in a recent metabolome study of 6136 Finnish men in the METSIM cohort measured by Metabolon platform[44]. This indicates that our novel associations replicated (at $P < 3e-3$) in an independent cohort from the same population using a different measurement technique. A missense variant in *SPHK2*, encoding sphingosine kinase 2 which plays an important role in sphingolipid metabolism, was found to be associated with SMs in our study and has also been reported for SMs by the METSIM study[44]. The unique LD pattern of the Finnish population also facilitated an identification of additional independent variants associated with lipids in the known lipid loci through fine-mapping. For example, the *LIPC* region has been reported to contain three independent signals for standard lipids[20] and in addition to these a fourth independent signal has been reported to be associated with lipid species[35]. In addition to these four signals, our study identifies a new independent signal at a missense variant (rs201563586), not in LD with any of the previously reported signals. This variant has a high PIP (>0.90) in our fine-mapping analysis and is highly enriched in the Finnish population, indicating the benefits of studying population isolates in genetic studies.

Another advantage of our study is the multivariate approach that showed a considerable gain in power in the discovery of new loci over the standard univariate GWAS. The multivariate GWAS identified 36% more BFS loci compared to univariate GWAS (55 vs. 35; Fig. 3) and discovered 6 new loci (*DTL*, *STK39*, *CDS1*, *YPEL2*, *KCNJ12*, and *AGPAT3*), not detected by the univariate GWAS. The interpretation of a multivariate association is often not straightforward in terms of the original traits. To improve the interpretation of the multivariate associations, we applied a recent statistical method[37] that decomposes the multivariate association into a smaller set of driver traits. Informative decompositions with at most 3 driver traits were observed for 78% of the BFS GWAS regions (108 of 138) found by the multivariate analysis. These regions represented eight such loci that did not reach BFS in any univariate analysis. Two examples are the association between cluster 7 and the *APOB* locus driven by CE 16:0;0 and CE 20:4;0, and the association between cluster 8 and a missense variant in *SPHK2* driven by SM 34:2;2 and SM 38:2;2.

Individual lipid species have been shown to predict cardiovascular disease risk more accurately than the standard lipids[21]. We observed disease associations with lipidome-associated variants for various disease groups (Fig. 4). For statin medication, we observed a shared genetic association with 58% of the lipid species and all multivariate clusters. Another widely lipidome-associated endpoint was cholelithiasis (47%). The multivariate clusters and CE species are sharing

genetic associations with all disease groups. Species of the classes SM, TAG, DAG, and a few species of the classes PC, PCO, PE, and PI show similar patterns for most disease groups, except for Alzheimer's disease, vascular dementia, or diabetic retinopathy. While these shared associations could point to interesting connections between lipid levels and diseases, there are two limitations with such observations. First, a shared association does not automatically mean that the potential causal variant in the region is the same for the lipid trait and the disease. However, the colocalization analysis supported that the causal variants are shared for most of the pairs of lipid species and diseases highlighted by the PheWAS analysis. Second, the disease endpoints in Fig. 4 have varying effective sample sizes and therefore some differences between the observed associations across the diseases could simply reflect the differences in statistical power.

We also examined the lipidomic profiles of 11 known CAD variants (Fig. 5). The CAD locus *ZNF259* showed the widest set of associations with 46 lipid species and 9 clusters of lipid species. The effect of the *ZNF259* polymorphism was previously only reported for standard lipids, with the first report[45] stating that individuals carrying the risk-increasing G allele showed increased TG levels and decreased LDL-C levels. We analyzed the effect of the polymorphism on lipid species: individuals with the G allele showed increased levels of DAGs, PCs, PE 18:0;0_18:2;0, PIs, and TAGs and decreased levels of PC O-16:1;0/18:1;0. Our list of marginal lipid associations of the CAD-associated variants contained three such CAD loci associated at GWS with lipid species or clusters that were not included in the previous report[35] of the CAD-associated lipid variants.

To summarize, our study identifies novel genetic loci with a role in lipid metabolism, points towards functional effects on detailed circulating lipid measures, and shows connections to cardiometabolic and related diseases. We also highlight the benefits of utilizing multivariate methods for association testing in multiple correlated phenotypes. Our comprehensive catalog of detailed lipid associations provides new opportunities for studying the role of lipids in disease-associated loci. For instance, the univariate GWAS summary statistics we provide could be utilized for development of polygenic risk scores and Mendelian Randomization studies. Further analyses employing group variables such as aggregating lipids sharing a particular fatty acid in their side chain or sharing a head group, or utilizing lipid ratios might lead to an identification of new associations.

## Methods
### Study participants

We use data from the prospective GeneRISK cohort[46] whose principal aim is to assess the impact of communication of genetic risk information of CVD to study participants. The cohort includes 7292 participants (4642 women, 2624 men), who were recruited from Southern Finland during 2015–2017 at age of 45–66 years. Participants were recruited from the Kymenlaakso province in South-Eastern Finland by identifying 4857 individuals from the population register at random and inviting them by mail. Further 1369 individuals were recruited from customers of Helsinki and Turku offices of a private and occupational health care provider. Additionally, online advertising was used to recruit 1116 blood donors. Individuals under guardianship, with previous history of Atherosclerotic Cardiovascular Disease and pregnant women were excluded from the study. The basic study characteristics are presented in Supplementary Table 3. The participants were instructed to fast overnight for 10 h before the collection of blood samples for plasma, serum, and DNA extraction. The biological samples (DNA, blood, serum, plasma) and the participants' demographic information and health data, genetic data and lipidomic data are stored in the THL Biobank [https://thl.fi/en/web/thl-biobank/for-researchers/sample-collections/generisk-study]. The GeneRISK study was carried out according to the principles of the Helsinki declaration and the Council of Europe's (COE) Convention of Human Rights and

Biomedicine. All study participants gave their informed consent to participate in the study. The study protocols were approved by The Hospital District of Helsinki and Uusimaa Coordinating Ethics committees (approval No. 281/13/03/00/14 (GeneRISK)).

Ethics statement for FinnGen is listed in Supplementary Note 3.

## Lipidomics

Mass spectrometry-based lipid analysis was performed for 7302 individuals from the GeneRISK cohort by shotgun lipidomic analysis at Lipotype GmbH (Dresden, Germany). The analysis was performed by direct infusion in a QExactive mass spectrometer from Thermo Scientific with a TriVersa NanoMate ion source from Advion Biosciences[47]. The lipidomics data were analyzed using lipid identification software and a data management system developed in-house by Lipotype GmbH[48,49]. The lipids with a high signal-to-noise ratio (>5) and amounts at least 5-fold higher than the corresponding blank samples were included. By including 8 reference samples per 96-well plate batch, reproducibility was assessed and the lipid amounts were corrected for batch variations and analytical drift if the $P$-value of the slope was <0.05 with an $R^2 > 0.75$ and the relative drift >5%. The lipid species detected in more than 70% of the samples were included (179 lipid species from 13 lipid classes). After the samples with a very low total lipid content and with >30% of the 179 lipids missing were excluded ($N = 26$), data from 7276 individuals remained.

The lipid molecules were identified at the species or subspecies level. The lipid species are named in the following notation: class name <sum of carbon atoms>:<sum of double bonds>;<sum of hydroxyl groups>. The annotation of lipid subspecies includes information on their acyl moieties and, if available, on their sn-position. The acyl chains are separated either by "_" if the sn-position on the glycerol cannot be resolved or else by "/". Further explanation is given by Gerl et al.[50]. The lipid identifiers of the SwissLipids database[51] [http://www.swisslipids.org] and the shorthand notation[52] are provided in Supplementary Data 1.

## Genotyping and imputation

Genotyping was performed using the HumanCoreExome BeadChip from Illumina Inc. (San Diego, CA, USA) and genotype calling was done with GenomeStudio and zCall at the Institute for Molecular Medicine Finland (FIMM). The genotype data was lifted over to the human genome build version 38 (GRCh38/hg38) according to the protocol described in [https://doi.org/10.17504/protocols.io.nqtddwn]. In the pre-imputation quality control (QC), potential outliers based on genetic ancestry were removed. We performed a principal component analysis (PCA) using 61,106 good quality (minor-allele frequency (MAF) ≥ 0.05, Hardy-Weinberg equilibrium $P$-value (HWE) > 1e-6 and missingness <10%) and approximately independent (LD pruning with PLINK v1.9: $r^2$ threshold of 0.2, window size 50 kb, step size 5) genetic variants. Based on the PCA and the place of birth information from the questionnaire, the individuals with non-Finnish ancestry or birthplace were removed. However, the samples born in Estonia, Russia, and Sweden, but clustered with the samples of Finnish ancestry in PCA, were retained in the analysis. Samples ($N = 30$) with extreme heterozygosity (beyond ±4 s.d) were excluded. After the quality control filtering, 7174 samples, consisting of 4579 females and 2595 males, with both genotype and lipidome data were considered for the subsequent analyses.

The genotype data pre-phasing was done with Eagle v2.3.5[53] with the number of conditioning haplotypes set to 20,000. The genotypes were imputed with Beagle v4.1[54] using the population-specific Sequencing Initiative Suomi (SISu) v3 reference panel based on high-coverage (25–30x) whole-genome sequences for 3775 Finnish individuals. The procedure is described in [https://doi.org/10.17504/protocols.io.nmndc5e]. In the post-imputation QC, the variants with imputation INFO score <0.70 and MAF < 0.01 were excluded and 12,776,997 variants remained. The measured levels of the lipid species

were adjusted for age, sex, collection site (clinic), lipid medication, first 10 genetic PCs, and ancestry (separate indicator variables for individuals born in Russia, Estonia, and Sweden) using linear regression. After adjusting for the above-mentioned covariates, the residuals were inverse-normal transformed and were used as outcome variables in the association analyses.

## Hierarchical clustering of lipid species

A hierarchical clustering was performed using the absolute pairwise Pearson correlations of the inverse-normal transformed plasma levels of lipids to identify clusters of correlated lipids for the multivariate GWAS. The clustering analysis was performed separately for glycerolipids (44 lipid species from TAGs and DAGs) and the remaining lipid species (135 species belonging to glycerophospholipid, sphingolipid, and sterol). As highly correlated traits cause instability in multivariate association analyses, we iteratively excluded one member from each pair of lipid species with a correlation >0.8 until no pair with a correlation >0.8 remained. The hierarchical clustering was performed on the remaining lipid species using an average Euclidean distance metric on the pairwise correlations and clusters were identified by visually inspecting the dendrogram.

We then calculated Variance inflation factors (VIF) within each cluster for each cluster member using the R package 'car'. (Technically, to apply the 'car' package, the cluster members were considered independent variables in a regression model where the outcome variable was a randomly generated variable whose exact value made no difference to the calculation of VIFs.) Through this approach, we identified cluster members that were highly correlated with some linear combination of the other members of the cluster. We iteratively removed the cluster member with the largest VIF until the maximum VIF within the cluster was below 5.

The hierarchical clustering of the absolute pairwise Pearson correlations led to 11 clusters of correlated lipid species (Supplementary Figs. 1 and 2). Based on the VIFs, one trait was removed from clusters 1 and 5 and two traits were removed from clusters 3 and 8. A heatmap of correlations for species included in the clusters is shown in Fig. 1. A list of lipid species included in each cluster before and after removing traits is given in Supplementary Data 1. Separate heatmaps of the correlations within each cluster are included in Supplementary Fig. 3 and the correlation values between the lipid species are listed in Supplementary Data 1.

We computed pairwise Pearson correlations between the 179 lipid species and the standard lipids (HDL-C, LDL-C, TC, and TG). The correlation values are shown in Supplementary Figs. 15 and 16 for the lipid species and lipid classes, respectively. The correlation values between the lipid species and the standard lipids are listed in Supplementary Data 1. For each lipid species, we obtained the maximum absolute correlation maxCor across the standard lipids. We compared the mean of these maxCor values between two groups of lipid species: (1) the lipid species associated with the loci not previously reported for the standard lipids and (2) the remaining lipid species.

## SNP-based heritability estimates

The SNP-based heritability estimates for each lipid species were calculated using biMM (release from 03.03.2017)[55]. The genetic relationship matrix (GRM) used for the heritability estimates was calculated using 849,501 LD-pruned autosomal SNPs with imputation INFO score >0.95, MAF > 0.01, and missingness <3%. The LD pruning was done with PLINK v1.9 using a window size of 1000 kb, step size of 1, and pairwise $r^2$ threshold of 0.7. Additionally, high LD regions were excluded[56].

## Univariate GWAS for 179 lipid species

The inverse-normal transformed residuals adjusted for the covariates mentioned above were used in the association analysis performed with

the linear-mixed-model software MMM v1.01[57]. The number of samples per GWAS ranged between 5287 and 7174 because the samples with missing values for a specific lipid species were excluded in the GWAS for that lipid species. After excluding very rare variants (MAF < 0.002) and the variants with low imputation quality (INFO < 0.8), 11,318,730 variants were included in the GWAS. To account for multiple testing, the Bonferroni-corrected significance (BFS) threshold was set as $P$-value < 7.352941e-10 (5e-8/68) as 68 principal components of the mean imputed lipidome data were required to explain >90% of the phenotypic variance. All $P$-values reported in this study are two-sided.

**Multivariate GWAS for 11 lipid clusters**
The multivariate analysis of the 11 clusters identified through the hierarchical clustering was performed with metaCCA v1.13.1[58].

Phenotypic correlations needed for the analysis were estimated from the GWAS summary statistics using metaCCA. The beta coefficients of the univariate GWAS were standardized using the formula suggested by metaCCA:

$$\beta_{stand} = \frac{\beta}{\sqrt{N}\,se} , \qquad (1)$$

where $N$ denotes the sample size of the respective univariate GWAS and se denotes the standard error. The metaCCA $P$-values were calculated from a chi-square distribution using the mean GWAS sample size as parameter $N$. The BFS threshold for the multivariate GWAS was set to $P$-value < 4.545455e-9 (5e-8/11) as the multivariate analysis was performed for 11 clusters.

The dataset used by Cichonska et al.[58] to test metaCCA consisted of variants with INFO > 0.8 and MAF > 0.05. To assess the robustness of the multivariate analysis for rare (MAF < 0.01) and low-frequency variants (0.01 ≤ MAF < 0.05) we simulated data under the null hypothesis for four SNPs with different MAFs covering range (0.005–0.042). In the simulation, the genotypes were permuted 100,000 times and then univariate GWAS were done with MMM and multivariate GWAS with metaCCA. The results of the simulation are described in detail in the Supplementary Note 4. We observed slightly inflated multivariate $P$-values for rare and low-frequency variants and are therefore correcting the multivariate $P$-values of such variants using the genomic inflation factor ($\lambda$)[59] determined through this simulation approach. The simulation was performed for each rare or low-frequency variant that reached the genome-wide significance level ($P$ < 5e-8) in the multivariate analysis but not in any of the univariate analyses.

Further, for each of the clusters, MetaPhat (release from 01.07.2020)[37] was applied to identify the traits driving the multivariate association at each lead variant of the multivariate analysis. The software determines sets of central traits for multivariate associations using Bayesian Information Criterion and $P$-value statistics. For each multivariate association, we report the driver traits and the optimal set of traits as defined by MetaPhat.

**Defining lead variants in the GWAS regions**
For both the univariate and multivariate GWAS, a lead variant in a GWAS was defined iteratively as the variant with the lowest $P$-value. After each new lead variant was identified, a 1.5 Mb region on each side of the variant defined the GWAS region of the lead variant, and other variants in that region were excluded from the further search for lead variants. For each GWAS, overlapping GWAS regions were combined into a single combined GWAS region, for which the lead variant is defined as the variant with the lowest $P$-value among the lead variants of the overlapping regions, and the other lead variants are listed as secondary lead variants. The maximum width of a region was set to 6 Mb, and for overlapping regions exceeding this threshold the original window size of ±1.5 Mb was iteratively shrunk by 10% until the width of the combined GWAS region was below 6 Mb (or the shrunk regions

did not overlap anymore). The process was stopped after no variant outside the GWAS regions had reached genome-wide significance (GWS) of $P$-value < 5e-8. Similarly, we also defined the lead variants that reached Bonferroni-corrected significance (BFS).

To determine which of the lead variants from the multivariate analysis were also identified by the univariate analyses, we checked whether there were such variants that reached BFS or GWS in the univariate GWAS of any trait included in the multivariate analysis and had $r^2$ > 0.1 with the lead variant of the multivariate analysis.

A lead variant was considered "novel" if the lead variant was not in LD ($r^2$ < 0.1) with any of the known variants identified in previous GWAS that included standard lipids or lipid species (listed in Supplementary Data 7). LD-proxies for the previously reported variants that were not included in our GWAS were obtained using LDproxy from LDlink release 5.3.3[60]. In LDproxy, we used the data on the 1000 Genomes project's Finnish population for LD calculation. For variants that were monoallelic in the Finnish reference panel, we did the LD calculation with the combined European population. From the SNPs with $r^2$ > 0.8 and within 500 kb of the target variant, the one with the highest $r^2$ was chosen as an LDproxy. For 448 variants, no proxy was found, of which 151 variants were monoallelic in both the Finnish and the European populations or were not biallelic variants, or were not contained in dbSNP build 155. For the remaining 297 of the 448 variants, none of the proxies were contained in our GWAS, no proxies with $r^2$ > 0.8 were found or the variants were not included in the 1000 G reference panel. For these 297 previously reported variants, we additionally checked if any of our lead variants were located within ±1.5 Mb.

We report the closest gene for all lead variants using SNP-nexus v4[61–65] (overlapped gene if available or nearest upstream or downstream gene). The variant's function was predicted with Variant Effect Predictor (VEP) v103.1 (McLaren et al. 2016) and the most severe function was annotated to the variant. The possible functions were ordered by severity according to the ranking from Ensembl [https://m.ensembl.org/info/genome/variation/prediction/predicted_data.html]. We defined the functional variants as having at least one of the following functions (ordered by severity from more severe to less severe): transcript_ablation, splice_acceptor_variant, splice_donor_variant, stop_gained, frameshift_variant, stop_lost, start_lost, transcript_amplification, inframe_insertion, inframe_deletion missense_variant, protein_altering_variant, splice_region_variant.

**GWAS of linear combination phenotypes (LCP-GWAS)**
To enable fine-mapping of the multivariate associations, linear combination phenotypes (LCP) were constructed as a weighted sum of the traits where the weights corresponded to the optimal combination phenotype reported by metaCCA for the lead variant[66]. GWAS region-specific LCP-GWAS were performed with fastGWA-mlm from GCTA v1.93.2[67], with the same covariates as were used in the univariate GWAS.

We calculated pairwise Pearson correlations between the LCP phenotypes and the standard lipids. (Supplementary Data 2).

**Fine-mapping**
Fine-mapping was performed with FINEMAP v1.4[68] for each GWAS region. For each fine-mapped region, the in-sample linkage disequilibrium (LD) matrix was computed using LDstore2[69] from the genotype dosages. The maximum number of causal variants in a locus was set to 10. The number of independent association signals for each fine-mapped GWAS region was determined by the number of informative credible sets (CS) among those CS for which FINEMAP gave the highest posterior probability. A CS was considered informative if the minimum $r^2$ among its variants was ≥ 0.1. We chose the top variant from each CS to represent the association signal except if the CS contained functional variants in high LD ($r^2$ > 0.95) with the top variant, in which case the functional variant having the largest $r^2$ with the top

variant was chosen as the representative variant[66]. The GWAS lead variant was chosen as the representative variant if no informative CS was obtained. The MHC region (chr 6: 25–34 Mb) was excluded from the fine-mapping and there the GWAS lead variant was defined as the representative variant.

## Defining independent signals and physical loci across all traits

Earlier we defined the GWAS regions separately in each univariate or multivariate GWAS and these regions were used in fine-mapping. Next, we used the representative variants from the fine-mapping results to determine the set of independent signals across all traits. We merged the signals across the traits if their representative variants were in LD ($r^2 \geq 0.1$). For each signal, we took the union of the corresponding GWAS regions to define physical boundaries for the signal and finally, we combined the overlapping signal regions to form a single set of physical loci across all traits. The locus definition process is summarized in a flow chart and visualized for the locus *LPL* in Supplementary Figs. 7 and 8, respectively. The locus names were defined by the closest gene to the top variant with the lowest *P*-value across the associated traits except if there was a missense variant among the top variants, in which case the locus was named by the gene corresponding to the missense variant. Novel loci are defined as loci containing only GWAS regions whose lead variants were all novel.

## Comparison of fine-mapping results to fine-mapping results of standard lipids

We checked how the variants that got a high posterior inclusion probability (PIP) > 0.9 in the GeneRISK data, or the other variants in the same locus in LD with them ($r^2 > 0.1$ in GeneRISK), were fine-mapped across the standard lipids (HDL-C, LDL-C, TG, TC) in the UK Biobank (UKB) data by Finucane lab [https://www.finucanelab.org/data]. For this, the chromosomal positions of the UKB data were lifted over to the human genome build version 38 (GRCh38/hg38) with liftOver[70]. We considered only the variants included in both data sets. We acknowledge that the UKB variants that were not present in GeneRISK, but that were in LD with a GeneRISK variant with a high PIP, could explain why some high PIP variants in GeneRISK may have lower PIP in UKB.

## Gene prioritization and pathway enrichment analysis

We prioritized genes for which we found functional variants with PIP > 0.5 in fine-mapping of the univariate or multivariate GWAS. For the functional variants, we obtained functional variant scores from Variant Annotation Integrator[71] and CADD scores from CADD v1.6[38].

Additionally, we performed a gene prioritization analysis using FOCUS v0.7[41], which computes credible sets of genes based on a posterior inclusion probability (PIP). We performed Transcriptome-wide association studies (TWAS) and tissue-agnostic fine-mapping with FOCUS using the GTEx v8 eQTL reference panel weight database and in-sample LD. We used the MASHR-based GTEx v8 eQTL databases from PrediXcan[72–74] to create the weight database. As input, we used the univariate GWAS and the multivariate LCP-GWAS filtered for INFO > 0.8 and MAF > 0.002 and cleaned the data with the munge command from FOCUS. We classified the GTEx tissues into two categories, category 1 containing subcutaneous adipose tissue, visceral adipose tissue, liver, and whole blood, which were deemed the most relevant for lipid-related phenotypes in a previous study[34], and category 2 containing the remaining tissues.

We utilized FUMA v1.3.7 software's GENE2FUNC tool[42] to obtain information on the expression of the prioritized genes and identify pathways enriched for the prioritized genes.

## Phenome-wide association analyses

Phenome-wide association analyses (PheWAS) were performed for the GWAS lead variants and the representative variants of credible sets in 377,277 participants from the FinnGen biobank (FinnGen release 9)[75].

From FinnGen, all 953 endpoints of the following categories (ICD-10 Chapter listed in parentheses if available) were included: cardiometabolic endpoints, diabetes endpoints, diseases marked as autoimmune origin, drug purchase endpoints, gastrointestinal endpoints, neoplasms from hospital discharge (II), neoplasms from cancer register (II), diseases of the blood and blood-forming organs and certain disorders involving the immune mechanism (III), endocrine nutritional and metabolic diseases (IV), diseases of the nervous system (VI), diseases of the circulatory system (IX), neurological endpoints, diseases of the digestive system (XI). These ICD-10 chapters were chosen because diseases within these chapters have been previously reported to be associated with changes in lipid metabolism, such as II: breast cancer[76], III: Systemic Lupus Erythematosus[77], IV: lipid metabolism disorders and diabetes mellitus[78], VI: Alzheimer's disease[79], IX: Coronary artery disease[35], XI: Nonalcoholic Fatty Liver Disease[80]. For all endpoints, at least 50 cases exist. The included endpoints for each data source are listed in Supplementary Data 5. We report the associated endpoints reaching the threshold $P < 0.05$ corrected for the number of included endpoints ($P < 0.05/953 = 5.24659e\text{-}5$) for each lead variant and representative variant of a credible set. Additionally, we identified the endpoints reaching the GWS threshold corrected for the number of included endpoints ($P < 5e\text{-}8/953 = 5.24659e\text{-}11$). Due to the high correlation between many endpoints, these thresholds are likely very stringent.

We focused on the PheWAS endpoints connected with >3 lipid species or multivariate clusters and then assigned disease groups to the endpoints. We selected 11 endpoints of 5 disease groups to be included in a heatmap. The selected endpoints were chosen by selecting the endpoint with the largest effective sample size $N_{\text{eff}}$ among the endpoints of the same disease and by selecting the endpoints with the most specific diagnoses based on expert medical knowledge. $N_{\text{eff}}$ was defined as:

$$N_{\text{eff}} = N\theta(1-\theta), \qquad (2)$$

with $\theta$ being the proportion of cases. We provide a list of the endpoints and their disease groups and effective sample size in Supplementary Data 5, where the selected endpoints are highlighted.

## Colocalization analysis

Our colocalization approach uses the probabilistic model from eCAVIAR[81] to integrate GWAS and eQTL data. The aim of the colocalization analysis is to find genomic regions where genetic association signals on two phenotypes colocalize to the same genetic variant(s). We base the colocalization analysis on the fine-mapping results of the phenotypes. Our own fine-mapping results were utilized for the univariate and multivariate lipid phenotypes. The FinnGen disease endpoints were fine-mapped with SuSiE[82]. The colocalization analysis was performed between the informative credible sets (minimum LD between variants $r^2 \geq 0.1$) of the univariate and multivariate lipid GWAS and the 953 Finngen endpoints utilized in the PheWAS analysis.

For each pair $(k, j)$ of credible sets $CS_{T,k}$ (from trait $T$) and $CS_{D,j}$ (from disease $D$), we compute the causal posterior probability (CLPP) as the the probability of a shared causal variant:

$$\text{CLPP}_{k,j} = \sum_{s \in CS_{T,k} \cap CS_{D,j}} \text{PIP}_{T,k}(s) \cdot \text{PIP}_{D,j}(s), \qquad (3)$$

where the sum is over the shared variants in both credible sets and the PIPs are the credible set-specific posterior inclusion probabilities. This CLPP calculation is similar to equation 8 in[81]. We used a CLPP threshold of 0.01 to suggest that the causal variants are shared as in[81].

## Association between coronary artery disease loci and lipidome

We assessed associations of the coronary artery disease (CAD) variants identified by a recent study[43] with the lipid species and clusters of lipid

species in our study. Of the 241 conditionally independent GWS associations with CAD at 198 loci, 236 variants at 196 loci were either included in our GWAS, or their LD-proxies were found in our GWAS (LD-proxies were defined using the same approach as with the lead variants). We summarized the associations at three levels of significance: (1) $P < 0.05$ corrected for multiple testing by the number of PCs explaining 90% of the variance (univariate analyses) or the number of clusters (multivariate analyses) ($P < 0.05/68 = 7.352941e\text{-}4$ for univariate and $P < 0.05/11 = 4.545455e\text{-}3$ for multivariate analyses), (2) the GWS threshold $P < 5e\text{-}8$ and (3) the GWS threshold corrected for multiple testing ($P < 5e\text{-}8/68 = 7.352941e\text{-}10$ for univariate and $P < 5e\text{-}8/11 = 4.545455e\text{-}9$ for multivariate analyses).

### Reporting summary

Further information on research design is available in the Nature Portfolio Reporting Summary linked to this article.

## Data availability

The univariate GWAS summary statistics generated in this study have been deposited in the GWAS catalog under accession codes GCST90277238-GCST90277416. The DNA, blood, serum, and plasma samples of the GeneRISK study participants, in addition to their demographic information, health, genotype, and lipidomics data are stored in the THL Biobank [https://thl.fi/en/web/thl-biobank/for-researchers/sample-collections/generisk-study]. The GeneRISK data are available under restricted access via procedures outlined in the Finnish Biobank Act and access can be obtained for biomedical research by contacting admin.biobank@thl.fi. A response to requests will be received within three weeks. Researchers may use the data only for purposes described in the application and are allowed to share the data with others only with a written approval from the THL Biobank.

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

## Acknowledgements

We would like to thank Johanna Aro, Sari Kivikko, and Ulla Tuomainen for management assistance in the project. We thank all study participants of the GeneRISK study for their participation. We acknowledge the participants and investigators of the FinnGen study. Full FinnGen acknowledgments and FinnGen funders are provided in the Supplementary Note 5. The GeneRISK study was funded by Business Finland through the Personalized Diagnostics and Care program coordinated by SalWe Ltd (grant No. 3986/31/2013). Dr Ripatti was supported by the Academy of Finland Center of Excellence in Complex Disease Genetics (grant No. 312062), the Finnish Foundation for Cardiovascular Research, the Sigrid Juselius Foundation, and University of Helsinki HiLIFE Fellow and Grand Challenge grants. Dr Pirinen was supported by the Academy of Finland (grants 338507 and 336825) and Sigrid Juselius Foundation.

## Author contributions

L.O., R.T., M.J.G., K.S., S.R., and M.P. conceived and designed the study; L.O. performed multivariate GWAS and all statistical analyses and reported the results; R.T. performed univariate GWAS; S.E.R. performed quality control of genotype data; L.O., R.T., M.J.G. E.W., K.S., S.R., and M.P. interpreted the results; M.J.G., C.K., and K.S. performed lipidomic profiling and processed the raw data; L.O. drafted the manuscript with help from R.T. and M.P; R.T., S.R., and M.P. supervised the study. All authors read, commented, and approved the manuscript.

## Competing interests

K.S. is CEO of Lipotype GmbH. K.S. and C.K. are shareholders of Lipotype GmbH. M.J.G. is employee of Lipotype GmbH. The remaining authors declare no competing interests.

## Additional information

## FinnGen

Sanni E. Ruotsalainen [1], Elisabeth Widén [1] & Samuli Ripatti [1,3,4]

A full list of members and their affiliations appears in the Supplementary Information.

