## [Peer Review File · Nature Communications]

Genome-wide association analysis of plasma lipidome identifies 495 genetic associationsREVIEWER COMMENTS

Reviewer #1 (Remarks to the Author):

The present study presents a GWAS with 179 lipidomics traits measured in 7,174 Finish individuals using the Lipotype platform. The authors report 495 associations at 56 loci.

This work is the largest GWAS done so far with this (or similar) lipidomics platform. Its associations represent a huge resource that will be of great interest to many researchers.

It should eventually be published by Nature Communications.

Having said this, I have several critical points.

The authors provide a multivariate analysis, using 11 variables that are derived from the original set by hierarchical clustering. I do not see the interest in this additional analysis. Other than adding a few hard to interpret associations, it is mostly technical. There is no true biochemical interpretation of the representative cluster variables, and I found the mentioning of the additional multi-variate associations throughout the paper mostly as a disturbance. I suggest removing this part entirely.

The authors may consider adding group variables instead, like the sum of all lipids have a given lipid X:Y in their side chain, or sharing a same head-group, or using principal components as dependent variables (in this case the same problem of interpretability as with using cluster variables arises, which would need to be addressed in either case). It is noteworthy that almost all cluster variables have higher inflation rates larger than the most inflated single lipid trait, which suggests that there are some issues with confounding when using cluster variables.

The authors conduct a heritability analysis. While many reviewers outside of the field request such analyses, I find them pointless when working with mass-spec derived metabolomics and lipidomics data, and this for two reasons: (1) in many cases, the heritability is driven by a few very strong lead variants, typically located in enzymes related to the molecule – in such cases heritability measures are just single associations summed up in a fuzzy way, and (2) heritability is defined as the part of the variance in a traits that can be explained by genetics – however, especially in mass-spec analyses, the non-genetic variability includes noise in the data, and that depends on and varies with the overall quantity of the molecule present. A lipid containing 16:0 or 18:0 fatty acids is always much more precisely measured than a rarer lipid, but this does not mean that these are more heritable. Therefore, heritability is expected to correlate with abundance, which makes it pointless. BTW – the authors found GWAS associations with molecules that have very low heritability. Again, personally, I would remove this part (but I acknowledge that other reviewers might insist on seeing it).

The analysis of causal variants and genes is not very informative and reads like a collection of applied methods rather than a quest for new insights. Again – and this is my personal opinion, I do not expect the authors to do it this way – I would drop all these algorithmic analyses and go through every single locus manually, identify the closest gene that is metabolically active, and then discuss its function in the context of the associated metabolites (not only the top association), as exemplified by Surendran et al.

(Nat Med 2022, PMID 36357675). Novelty is not necessarily limited to the genetic variant, but to the entire associated metabotype (or should I say lipotype here?). If a variant is associated at a genome- and lipidomewide level with one trait, then it is reasonable to argue that the genetic association is not found by chance, and hence do a follow-up analysis in which to test only the 56 loci found in this study with all traits, use a more lenient p-value threshold of $p < 0.05 / 179 / 56$, and then consider all lipids that meet this criterion as part of an associated lipotype. On these associations I would personally conduct a joint local lipotype and genotype refinement, as demonstrated in Fig. 4 of Surendran et al.

A flagrant example of a missed annotation is 11:61785208 in table 1: this is rs174537 at the FADS locus, one of the most prominent lipidomics loci in the human genome. In table 1 it is annotated as MYRF (and the p-value is approximated as $<5E-324$, probably because the authors did not work with log-scaled p-values, which should be done when working in a GWAS of this scale). And please include rs identifiers in table 1 – it makes cross-referencing other studies so much easier.

Data availability: Please confirm that “Univariate GWAS summary statistics will be available on GWAS catalog” will include the full summary statistics, not just the lead hits, so that the data could be used for Mendelian Randomization studies, polygenic risk score development, etc. This would add value to this study as a resource.

Minor points

Figure 4 is not very readable

Table 3: replace “Inf” by “population specific (I assume these variants were not detected outside Finland???)”

Check the abstract and text for English. Articles are missing in some places (e.g. “Human plasma lipidome captures risk for cardio-metabolic diseases.”)

Reviewer #2 (Remarks to the Author):

The authors report a large-scale univariate and multivariate GWAS of 179 lipid species in 7,174 Finnish individuals, followed by fine-mapping, gene prioritization and PheWas.

Generally, this is a good quality, well written manuscript. However, I have some questions/concerns/suggestions

MAJOR

1- Results: first paragraph: “Hierarchical clustering based on absolute pairwise Pearson correlations of plasma levels of lipids revealed 11 clusters of correlated lipids”.

It seems like clustering was all based on Pearson correlations which assumes normality, linearity, and homoscedasticity, were these assumptions satisfied in the lipid species? usually lipid species and metabolites in general are not normally distributed and, in this manuscript, the authors used inverse normal transformation to perform the association analysis. So I wonder if Spearman correlation would be more appropriate and how much that might affect clustering

2- Heritability of lipid species: How do these estimates compare to other previously reported SNP/pedigree-based estimates in similar/different populations? This should be discussed.

3- Methods: Defining lead variants in the GWAS regions: “A lead variant was considered “novel” if the lead variant was not in LD ($r^2 < 0.1$) with any of the known variants identified in previous GWAS that included standard lipids or lipid species (listed in Supplementary Data 5).”

The list for lipid species includes 14 previously published GWAS but does not include what I think is the most appropriate one (Nature Communications volume 13, Article number: 1644 (2022), PMID: 35347128), as it used 6,136 Finnish men (METSIM). While the METSIM study used all metabolomics and not just lipidomics, their set of 1,391 metabolites includes 548 lipid species, so it would be interesting to compare results between these two studies.

In checking the 9 novel loci in Table 2 using the METSIM results in the pheweb (<https://pheweb.org/metsim-metab/>), I found 8 of the 9 loci having association with one or more lipid species with p values ranging from $E-03$ to $E-06$. That does not take away the novelty from this manuscript since these p values are not considered significant, however this is actually considered a good significant replication for these 8 loci and should be reported as a supported information for these findings. It's usually challenging to find a replication cohort for isolated population, but in this particular case there is a source for replication with a descent sample size that used a different technique for profiling, so using it for replication is ideal.

For one variant (19 : 48,629,610 G / C (rs61751862)) in SPHK2, there was a significant association ($3.1e-11$) with probably the same sphingomyelin (d18:1/20:1, d18:2/20:0).

4- The value of this manuscript can significantly enhanced by performing colocalization analysis to determine if the lipid species and disease/trait share the same variants. And also to determine directionality/causality using bidirectional Mendelian Randomization. While it used to be challenging to perform MR with omic data due to pleiotropy, currently there are multiple options for multivariable MR designed to address this issue. These results can also be very useful and informative to the field given the relatively large sample size and the unique isolated Finnish population.

5- The authors may also consider investigating lipid ratios (PMID: 34503513, PMID: 36635386), as we may get unique insight from this special population, and you may find some interesting novel results that you can discuss in detail.

MINOR

- Results: first paragraph: the first referenced Supplementary Table is Table 2, and the first referenced Supplementary Figure is Fig 10-11. I'm not sure about the policy of the journal but I think it would be better to reorder the Supplementary tables and figures as they appear in the text.

- Figure 3: it's a little hard to distinguish the grid line from the significant lines, maybe use lighter/darker shades or different pattern/colors

- Coronary artery disease loci associations: Please clarify what is FinnGen R9

- Figure 4: the order of clusters is 1,10,11,2,...it's better to be 1,2,3...

- Discussion: second paragraph: "Here, we have identified 15 lipid-associated loci that were not captured even by the largest GWAS of standard lipids with >1.65 million participants." Is this a typo? Throughout the manuscript there were 9 novel loci not 15?

Response to Reviewers' comments

We thank the editor and the reviewers for their valuable comments which helped to improve the manuscript substantially. We provide the point-by-point response to each comment below, along with the text modifications made in the revised manuscript based on the comment. The major changes are highlighted in red font in the revised manuscript.

Reviewer #1 (Remarks to the Author):

The present study presents a GWAS with 179 lipidomics traits measured in 7,174 Finish individuals using the Lipotype platform. The authors report 495 associations at 56 loci. This work is the largest GWAS done so far with this (or similar) lipidomics platform. Its associations represent a huge resource that will be of great interest to many researchers. It should eventually be published by Nature Communications. Having said this, I have several critical points.

1. The authors provide a multivariate analysis, using 11 variables that are derived from the original set by hierarchical clustering. I do not see the interest in this additional analysis. Other than adding a few hard to interpret associations, it is mostly technical. There is no true biochemical interpretation of the representative cluster variables, and I found the mentioning of the additional multi-variate associations throughout the paper mostly as a disturbance. I suggest removing this part entirely.

We thank the reviewer for the valuable comments and suggestions. Multivariate analysis of multi-dimensional data provides higher statistical power and thus enables identification of new signals, as also demonstrated by our study. Using multivariate analysis, we identified 20 additional loci which were not found by univariate analysis (55 vs. 35 loci). Moreover, the multivariate analyses provided statistically clearer associations (lower P-values) for 47 out of 55 loci (Table 1 and Figure 3) even when we do not penalize the univariate tests by their larger multiple testing burden. Thus, we believe that multivariate analyses provide additional value in terms of identification of new and robust association signals, and therefore we prefer to include these results in the manuscript.

We agree with the reviewer that the biochemical interpretations of associations of the multivariate analyses are not straightforward. To help this interpretation, we have traced down the multivariate associations to smaller sets of driver traits (lipid species driving each multivariate association). Among the 55 loci, there were in total 138 separate multivariate associations and in 61 of them, we were able to trace down the source of the association to a single driver trait. For a large majority of the multivariate associations (108 / 138), there were at most 3 driver traits that jointly were accountable for that signal. We report these driver traits in Supplementary Table 6 and explain our analysis approach based on the MetaPhat software on p.18.

We have modified the following text in the Results (page 6) and Discussion sections (page 14), respectively:

”Further, **to improve the interpretation of multivariate associations** we applied MetaPhat [37] to identify the **lipid species** driving the multivariate association. For 61 of all 138 multivariate BFS GWAS regions, a single **lipid species** was identified to be driving the association and for 47 of the regions, 2 to 3 driver traits were identified. The driver traits and other MetaPhat results for the associations reaching GWS in the multivariate analysis are listed in Supplementary Table 6.”

”Another advantage of our study is the multivariate approach that showed a considerable gain in power in the discovery of new loci over the standard univariate GWAS. The multivariate GWAS identified 36% more BFS loci compared to univariate GWAS (55 vs. 35; Figure 3) and discovered 6 new loci (*DTL*, *STK39*, *CDS1*, *YPEL2*, *KCNJ12*, and *AGPAT3*), not detected by the univariate GWAS. The interpretation of a multivariate association is often not straightforward in terms of the original

traits. **To improve the interpretation of the multivariate associations**, we applied a recent statistical method [37] that decomposes the multivariate association into a smaller set of driver traits. Informative decompositions with at most 3 driver traits were observed for 78% of the BFS GWAS regions (108 of 138) found by the multivariate analysis. These regions represented eight such loci that did not reach BFS in any univariate analysis. Two examples are the association between cluster 7 and the *APOB* locus driven by CE 16:0;0 and CE 20:4;0, and the association between cluster 8 and a missense variant in *SPHK2* driven by SM 34:2;2 and SM 38:2;2.”

2. The authors may consider adding group variables instead, like the sum of all lipids have a given lipid X:Y in their side chain, or sharing a same head-group, or using principal components as dependent variables (in this case the same problem of interpretability as with using cluster variables arises, which would need to be addressed in either case).

We thank the reviewer for the suggestion of adding group variables to study further characteristics of the lipid species. Considering the scope and amount of data already in the current manuscript, we feel such analyses focusing on a larger set of group variables and lipid ratios (as suggested by Reviewer 2) should be performed and presented in a separate study allowing a more focussed interpretation. We have now discussed this potential analysis in the Discussion section (page 15):

”Further analyses employing group variables such as aggregating lipids sharing a particular fatty acid in their side chain or sharing a head group, or utilizing lipid ratios might lead to an identification of new associations.”

3. It is noteworthy that almost all cluster variables have higher inflation rates larger than the most inflated single lipid trait, which suggests that there are some issues with confounding when using cluster variables.

The question of possible inflation of multivariate tests due to confounding or technical problems is indeed very important and we have studied this in detail on p.1-2 of Supplementary Note. There, we performed a simulation analysis and found that multivariate *P*-values for rare and low-frequency variants ($MAF < 0.05$) are slightly inflated but that multivariate *P*-values for common variants ($MAF > 0.05$) are not inflated. Out of our multivariate lead variants that did not reach $P < 5e-8$ in any univariate GWAS, only 5 had $MAF < 0.05$, and we have corrected their multivariate *P*-values (reported in Supplementary Note Table 1) using variant-specific genomic inflation factors determined through the simulation approach. We explain this approach on p.1 of Supplementary Note. Additionally, we would like to remind that since the statistical power seems larger in the multivariate analyses than in univariate analyses (Figure 3), we expect that this power difference is also reflected in the higher inflation values in the multivariate analyses than in the univariate analyses even without any confounding.

4. The authors conduct a heritability analysis. While many reviewers outside of the field request such analyses, I find them pointless when working with mass-spec derived metabolomics and lipidomics data, and this for two reasons: (1) in many cases, the heritability is driven by a few very strong lead variants, typically located in enzymes related to the molecule – in such cases heritability measures are just single associations summed up in a fuzzy way, and (2) heritability is defined as the part of the variance in a traits that can be explained by genetics – however, especially in mass-spec analyses, the non-genetic variability includes noise in the data, and that depends on and varies with the overall quantity of the molecule present. A lipid containing 16:0 or 18:0 fatty acids is always much more precisely measured than a rarer lipid, but this does not mean that these are more heritable. Therefore, heritability is expected to correlate with abundance, which makes it pointless. BTW – the authors found GWAS associations with molecules that have very low heritability. Again, personally, I would remove this part (but I acknowledge that other reviewers might insist on seeing it).

We thank the reviewer for the insightful comments about heritability estimates of lipid species. We now acknowledge this issue in the discussion. We still prefer to keep the heritability estimates in the manuscript as the heritability analysis has been included in most lipidomics GWAS papers, and the readers might be interested to compare with other studies. As requested by Reviewer 2, we have also included a comparison to heritability estimates reported by other studies.

We have now added a comment about this in the Discussion (page 13):

“We acknowledge that the differences in heritability estimates between the lipid species may reflect also the differences in measurement accuracy in addition to the differences in the actual heritabilities.”

5. The analysis of causal variants and genes is not very informative and reads like a collection of applied methods rather than a quest for new insights. Again – and this is my personal opinion, I do not expect the authors to do it this way – I would drop all these algorithmic analyses and go through every single locus manually, identify the closest gene that is metabolically active, and then discuss its function in the context of the associated metabolites (not only the top association), as exemplified by Surendran et al. (Nat Med 2022, PMID 36357675). Novelty is not necessarily limited to the genetic variant, but to the entire associated metabolite (or should I say lipotype here?). If a variant is associated at a genome- and lipidomewide level with one trait, then it is reasonable to argue that the genetic association is not found by chance, and hence do a follow-up analysis in which to test only the 56 loci found in this study with all traits, use a more lenient p-value threshold of $p < 0.05 / 179 / 56$, and then consider all lipids that meet this criterion as part of an associated lipotype. On these associations I would personally conduct a joint local lipotype and genotype refinement, as demonstrated in Fig. 4 of Surendran et al.

We thank the reviewer for these suggestions. In this work, we chose to perform algorithmic analyses to identify causal variants and genes in order to avoid subjective choices and to keep our results replicable. We have now performed a follow-up analysis where we tested the associations of all lead variants and representative variants with all lipid species. We provide a table of GWAS summary statistics for all 179 lipid species for all lead or representative variants found for a univariate or multivariate trait in Supplementary Data 2. We have added the following text to the Results section (page 5):

“We provide the GWAS summary statistics for all 330 lead variants and representative variants for all 179 lipid species (Supplementary Data 2). Of the 59,070 variant-lipid associations, 7,814 reached the marginal significance threshold of $P < 0.05/85/68=4.11e-6$ corrected for the 85 GWS loci and the 68 PCs that together explained 90% of the phenotypic variance.”

6. A flagrant example of a missed annotation is 11:61785208 in table 1: this is rs174537 at the FADS locus, one of the most prominent lipidomics loci in the human genome. In table 1 it is annotated as MYRF (and the p-value is approximated as $<5E-324$, probably because the authors did not work with log-scaled p-values, which should be done when working in a GWAS of this scale). And please include rs identifiers in table 1 – it makes cross-referencing other studies so much easier.

We thank the reviewer for these detailed observations. We have now added the rsids to Table 1 to allow easy cross-referencing and the rsids are available for all variants in Supplementary Data 2. We have also computed the *P*-values for the *FADS* region using the logarithmic-scale. As a consequence, the lead variant has now changed to rs174562 and the name of the locus to *FADS2* since it is the gene nearest to the new lead variant.

7. Data availability: Please confirm that “Univariate GWAS summary statistics will be available

on GWAS catalog” will include the full summary statistics, not just the lead hits, so that the data could be used for Mendelian Randomization studies, polygenic risk score development, etc. This would add value to this study as a resource.

We agree with the reviewer and confirm that the full summary statistics will be made available to the GWAS catalog for future studies. We have added the following text to the Discussion (page 15).

“For instance, the univariate GWAS summary statistics we provide could be utilized for development of polygenic risk scores and Mendelian Randomization studies.”

Minor points

Figure 4 is not very readable

We have improved the readability of the Figure by increasing the size of axis marks, by increasing the width of the figure and by adding gaps between the rows.

Table 3: replace “Inf” by “population specific (I assume these variants were not detected outside Finland???)”

Yes, the variants were not detected outside Finland. We have replaced Inf with “FIN-specific” and clarified this in the Table description.

Check the abstract and text for English. Articles are missing in some places (e.g. “Human plasma lipidome captures risk for cardio-metabolic diseases.”)

We have checked the abstract and manuscript text and fixed grammatical errors and missed articles.

Reviewer #2 (Remarks to the Author):

The authors report a large-scale univariate and multivariate GWAS of 179 lipid species in 7,174 Finnish individuals, followed by fine-mapping, gene prioritization and PheWas. Generally, this is a good quality, well written manuscript. However, I have some questions/concerns/suggestions

MAJOR

1. Results: first paragraph: “Hierarchical clustering based on absolute pairwise Pearson correlations of plasma levels of lipids revealed 11 clusters of correlated lipids”. It seems like clustering was all based on Pearson correlations which assumes normality, linearity, and homoscedasticity, were these assumptions satisfied in the lipid species? usually lipid species and metabolites in general are not normally distributed and, in this manuscript, the authors used inverse normal transformation to perform the association analysis. So I wonder if Spearman correlation would be more appropriate and how much that might affect clustering

We thank the reviewer for pointing this out. We have computed Pearson correlation on the phenotypes after the inverse normal transformation and thereby assured that the assumptions of normality and homoscedasticity are satisfied. We have clarified this by modifying the following text in the Methods section (page 17):

”A hierarchical clustering was performed using the absolute pairwise Pearson correlations of the inverse normal transformed plasma levels of lipids to identify clusters of correlated lipids for the multivariate GWAS.”

To see the difference between Spearman and Pearson correlation, we have calculated Spearman correlations and compared them to Pearson correlations. Spearman correlations provided similar

clusters with only minor differences as shown in the figure below. We therefore chose to keep Pearson correlations for the hierarchical clustering. The figure below compares Spearman correlations to Pearson correlations for the 179 lipid species.

Absolute values of Spearman correlations (upper triangle) and Pearson correlations (lower triangle) of inverse normal transformed plasma levels of lipids. Lipid species are ordered in alphabetical order as in Supplementary Table 1.

2. Heritability of lipid species: How do these estimates compare to other previously reported SNP/pedigree-based estimates in similar/different populations? This should be discussed.

We agree that it is important to compare the heritability estimates to those reported by other studies. We added the following text to the Discussion (page 13):

“We found that the heritability estimates of lipid species ranged from 0 to 0.45. Previous studies have reported heritability estimates ranging from 0.10 to 0.54 [21], [35]. We observed the highest median heritability estimates for the lipid classes SM, and Cer (>0.30). A previous Finnish study reported the highest heritability estimates for Cer (0.39) [21]. In a recent Australian study [35] among the lipid classes included in our study, CE and LPE reached the largest heritability (0.38) and also Cer (0.34) and SM (0.36) were among the most heritable classes. We acknowledge that the differences in

heritability estimates between the lipid species may reflect also the differences in measurement accuracy in addition to the differences in the actual heritabilities.”

3. Methods: Defining lead variants in the GWAS regions: “A lead variant was considered “novel” if the lead variant was not in LD ($r^2 < 0.1$) with any of the known variants identified in previous GWAS that included standard lipids or lipid species (listed in Supplementary Data 5).”

The list for lipid species includes 14 previously published GWAS but does not include what I think is the most appropriate one (Nature Communications volume 13, Article number: 1644 (2022), PMID: 35347128), as it used 6,136 Finnish men (METSIM). While the METSIM study used all metabolomics and not just lipidomics, their set of 1,391 metabolites includes 548 lipid species, so it would be interesting to compare results between these two studies.

In checking the 9 novel loci in Table 2 using the METSIM results in the pheweb (<https://pheweb.org/metsim-metab/>), I found 8 of the 9 loci having association with one or more lipid species with p values ranging from $E-03$ to $E-06$. That does not take away the novelty from this manuscript since these p values are not considered significant, however this is actually considered a good significant replication for these 8 loci and should be reported as a supported information for these findings. It’s usually challenging to find a replication cohort for isolated population, but in this particular case there is a source for replication with a descent sample size that used a different technique for profiling, so using it for replication is ideal.

For one variant (19 : 48,629,610 G / C (rs61751862)) in *SPHK2*, there was a significant association ($3.1e-11$) with probably the same sphingomyelin (d18:1/20:1, d18:2/20:0).

We thank the reviewer for pointing out the METSIM study and agree that it should now be included in the list of previous GWAS. We have added it to the comparison with the previous studies listed in Supplementary Data 6, which changed the status of the locus *SPHK2* to a previously known locus. During this analysis, we also added 1176 more variants from the study by Graham et al. from 2021 among the previously known HDL, LDL, and TC variants that were missing from the previous version of our manuscript. We added the following text to the Discussion (page 13):

“The Finnish genetic background of our study population provides a unique opportunity to discover variants that are enriched in the Finnish population but extremely rare outside of Finland, and to identify new independent signals in known lipid loci. We identified two new lipid-associated loci that are enriched in the Finnish population. **These two enriched loci and the other 6 new loci reach P -values between $9e-6$ and $2e-3$ for lipids in a recent metabolome study of 6,136 Finnish men in the METSIM cohort measured by Metabolon platform [44]. This indicates that our novel associations replicated (at $P < 3e-3$) in an independent cohort from the same population using a different measurement technique. A missense variant in *SPHK2*, encoding sphingosine kinase 2 which plays an important role in sphingolipid metabolism, was found to be associated with SMs in our study and has also been reported for SMs by the METSIM study [44].”**

4- The value of this manuscript can significantly enhanced by performing colocalization analysis to determine if the lipid species and disease/trait share the same variants. And also to determine directionality/causality using bidirectional Mendelian Randomization. While it used to be challenging to perform MR with omic data due to pleiotropy, currently there are multiple options for multivariable MR designed to address this issue. These results can also be very useful and informative to the field given the relatively large sample size and the unique isolated Finnish population.

We thank the reviewer for these analysis suggestions. We agree that colocalization analysis will enhance the value of the manuscript and we have now added colocalization analysis between the univariate and multivariate associations and the disease endpoints, as explained on p.21 of the

Methods section. We have added the following text to the Results section (page 11) and modified the Discussion (page 14), respectively:

“Figure 4 shows the connection between 9 selected PheWAS endpoints (representing cardiovascular disease, hyperlipidemia, diabetes, metabolic disorders, and neurological disease) and the lipid species and multivariate clusters through shared associated variants. Only the associations reaching the GWS threshold corrected for the number of endpoints ($P < 5.25e-11 = 5e-8/953$) and simultaneously showing a colocalization ($CLPP > 0.01$) are illustrated. Of the 179 lipid species, 137 species are included in Figure 4. We identified 45 instances where links in the loci *FADS2*, *ZPR1*, *CERS4*, *TM6SF2*, and *HNF1A* were detected in PheWAS but not by the colocalization analysis, and they together with the colocalization results for all 953 FinnGen endpoints can be found in Supplementary Data 5. Supplementary Figure 12 shows the connection of all 45 BFS endpoints, ordered by disease groups, connected with > 3 lipid species.”

“While these shared associations could point to interesting connections between lipid levels and diseases, there are two limitations with such observations. First, a shared association does not automatically mean that the potential causal variant in the region is the same for the lipid trait and the disease. However, the colocalization analysis supported that the causal variants are shared for most of the pairs of lipid species and diseases highlighted by the PheWAS analysis. Second, the disease endpoints in Figure 4 have varying effective sample sizes and therefore some differences between the observed associations across the diseases could simply reflect the differences in statistical power.”

We agree that Mendelian randomization (MR) analyses would be interesting to determine directionality and possible causality. However, they require disease GWAS from large meta-analyses to be informative. Luckily, such data sets have been collected in the MR-Base platform that can carry out MR analyses between pairs of traits. Thus, a natural way to carry out MR analyses with our traits would be to link our GWAS summary statistics with the existing MR-Base platform. In order to make this possible, we will provide the GWAS summary statistics through GWAS catalog, which will allow inclusion of these GWAS summary statistics into future updates of MR-Base. We have added the following text to the Discussion (page 15):

“Our comprehensive catalog of detailed lipid associations provides new opportunities for studying the role of lipids in disease-associated loci. For instance, the univariate GWAS summary statistics we provide could be utilized for development of polygenic risk scores and Mendelian Randomization studies.”

5. The authors may also consider investigating lipid ratios (PMID: 34503513, PMID: 36635386), as we may get unique insight from this special population, and you may find some interesting novel results that you can discuss in detail.

We thank the reviewer for the suggestion of investigating lipid ratios. Considering the scope and the amount of data in the current manuscript, we feel such analyses focusing on a large number of derived traits, such as lipid ratios, should be performed and presented in a separate study to allow a more focussed interpretation.

We have added the following text to the Discussion section (page 15):

“Further analyses employing group variables such as aggregating lipids sharing a particular fatty acid in their side chain or sharing a head group, or utilizing lipid ratios might lead to an identification of new associations.”

MINOR

- Results: first paragraph: the first referenced Supplementary Table is Table 2, and the first referenced Supplementary Figure is Fig 10-11. I'm not sure about the policy of the journal but I think it would be better to reorder the Supplementary tables and figures as they appear in the text.

We have checked the numbering of Supplementary Tables and Figures and ordered them as they appear in the text.

- Figure 3: it's a little hard to distinguish the grid line from the significant lines, maybe use lighter/darker shades or different pattern/colors

We have improved the figure by choosing different colors and line styles.

- Coronary artery disease loci associations: Please clarify what is FinnGen R9

We have clarified in the text that FinnGen R9 stands for FinnGen Release 9.

- Figure 4: the order of clusters is 1,10,11,2,...it's better to be 1,2,3...

We have changed the ordering of clusters in Figure 4 and also in Supplementary Figure 12.

- Discussion: second paragraph: "Here, we have identified 15 lipid-associated loci that were not captured even by the largest GWAS of standard lipids with >1.65 million participants." Is this a typo? Throughout the manuscript there were 9 novel loci not 15?

In this paragraph, we are talking about the loci not reported by the existing GWAS of the standard lipids. Thus, these 15 loci include loci that were previously reported by lipidome studies but not by the GWAS of the standard lipids. We have modified the text on page 13 to clarify this:

"Here, we have identified 13 lipid-associated loci that were not captured **by any standard lipids study**, not even by the largest GWAS of standard lipids with >1.65 million participants [20]."

REVIEWERS' COMMENTS

Reviewer #1 (Remarks to the Author):

The author have responded to all relevant previous concerns.

Reviewer #2 (Remarks to the Author):

I'm satisfied with the author's response and have no further comments.